# QeRL: Quantization-enhanced Low-Rank Reinforcement Learning for LLMs

Wei Huang[1,3]   Yi Ge[2,4]   Shuai Yang[1]   Yicheng Xiao[4]   Huizi Mao[1]   Yujun Lin[1]

Hanrong Ye[1]   Sifei Liu[1]   Ka Chun Cheung[1]   Hongxu Yin[1]   Yao Lu[1]

Xiaojuan Qi[3]   Song Han[1,2]   Yukang Chen[1]

[1]NVIDIA   [2]MIT   [3]HKU   [4]THU

https://github.com/NVlabs/QeRL

## Abstract

We propose QeRL, a Quantization-enhanced Reinforcement Learning framework for large language models (LLMs). While RL is essential for LLMs' reasoning capabilities, it is resource-intensive, requiring substantial GPU memory and long rollout durations. QeRL addresses these issues by combining NVFP4 quantization with Low-Rank Adaptation (LoRA), accelerating rollout phase of RL while reducing memory overhead. Beyond efficiency, our findings show that quantization noise increases policy entropy, enhancing exploration in LoRA-based RL, and enabling the discovery of better strategies during RL. QeRL further introduces an Adaptive Quantization Noise (AQN) mechanism, which dynamically adjusts noise during training. Experiments demonstrate that QeRL delivers over $1.5\times$ speedup in the rollout phase compared to QLoRA, and around $1.3\times$ speedup compared to BF16 LoRA in 7B model. Moreover, this is the first framework to enable RL training of a 32B LLM on a single H100 80GB GPU, while delivering overall speedups for RL training. It also achieves faster reward growth and higher final accuracy than 16-bit LoRA and QLoRA, while matching the performance of full-parameter fine-tuning on mathematical benchmarks such as GSM8K (90.8%) and MATH 500 (77.4%) in the 7B model. These results establish QeRL as an efficient and effective framework for RL training in LLMs.

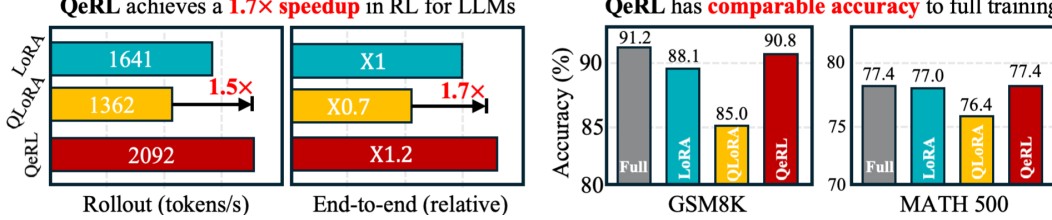

Figure 1: Rollout speedup and accuracy of QeRL on Qwen2.5-7B-Instruct. QeRL achieves faster RL rollout and end-to-end training speeds (batch=8), while delivering performance superior to vanilla LoRA and QLoRA, also comparable to full-parameter RL on mathematical benchmarks.

## 1  Introduction

The ability to perform multi-step reasoning is critical for large language models (LLMs) to handle complex tasks, from theoretical problem solving to practical decision making (Sui et al., 2025; Xu et al., 2025; Chu et al., 2025; Yang et al., 2021). Supervised fine-tuning (SFT) is a common method to improve reasoning by training models to replicate explicit reasoning steps (Huang et al., 2024d; Min et al., 2024). However, this approach risks promoting imitation rather than encouraging genuine reasoning. In contrast, reinforcement learning (RL) uses verifiable reward signals to support adaptive learning, allowing models to explore diverse reasoning traces and identify more robust solutions (Lambert et al., 2024; DeepSeek-AI, 2025; Chen et al., 2025a).

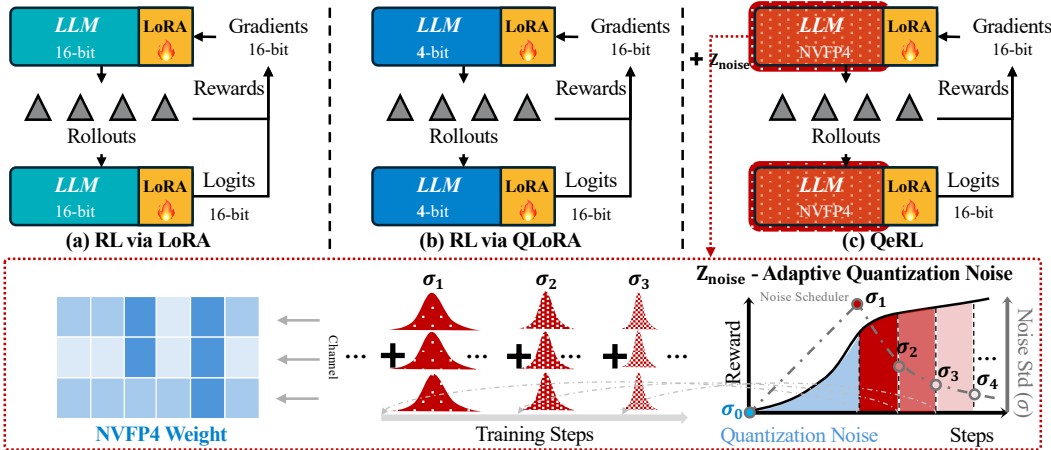

Figure 2: The illustration of QeRL. (a) **RL via LoRA**: reducing trainable parameters, but does not alleviate the rollout bottleneck. (b) **RL via QLoRA**: NF4 quantization with LoRA, but NF4 is slower than LoRA. (c) **QeRL**: NVFP4 quantization with LoRA, reducing memory and enabling faster RL while matching full-parameter finetuning performance with adaptive quantization noise. AQN dynamically adjusts quantization noise, enhancing exploration in LoRA-based RL.

RL is effective for LLMs' reasoning but highly resource-intensive. RL requires substantial GPU memory, as multiple models, such as policy and reference models in GRPO (Shao et al., 2024), must run concurrently. The large size of reasoning-focused LLMs (DeepSeek-AI, 2025) further exacerbates memory demands. Training is also slowed by multistage processes, including rollouts, reward computation, logit evaluation, and gradient updates. Rollouts are particularly costly, involving repeated sampling and processing of long sequences for complex tasks (Yu et al., 2025). Additionally, RL's inherent sample inefficiency (Hassani et al., 2024) further increases costs.

Improving RL efficiency in LLMs presents significant challenges. One approach, exemplified by Tina (Wang et al., 2025), leverages parameter-efficient fine-tuning methods like Low-Rank Adaptation (LoRA) (Hu et al., 2022) to reduce trainable parameters. However, similar to LoRA in SFT (Chen et al., 2024c), these methods fail to address the core issue of slow rollout speeds. Another strategy, demonstrated by FlashRL (Liu et al., 2025a), uses quantized rollout models to reduce computational costs. However, precision mismatches between the rollout model and logits model (e.g., 8-bit vs. 16-bit) require importance sampling to correct discrepancies, necessitating both 8-bit and 16-bit models to run simultaneously, which increases memory usage. This usage pattern is conceptually similar to QuaRL (Krishnan et al., 2019) (proposed for traditional RL task of small model), where a quantized model is used for rollouts while full-precision weights are still needed for policy optimization; as a result, the train–inference mismatch (especially pronounced for LLMs) persists and the training-time memory footprint remains close to that of full-parameter fine-tuning. To overcome these limitations, we focus on lower-bit quantization while avoiding duplicate models in memory. Additionally, using QLoRA (Dettmers et al., 2023a) in RL slows rollouts by 1.5–2×, further reducing efficiency. This slowdown occurs because QLoRA relies on NormalFloat 4-bit (NF4) precision, which requires unpacking and mapping to floating-point values via a lookup table before matrix multiplication.

To address the limitations of NF4 in QLoRA, a natural solution is to adopt higher-performance quantization. However, standard quantization methods introduce static and deterministic noise, which is non-beneficial to the later-stage RL training. To avoid this drawback, our analysis surprisingly reveals that quantization noise, with precise control, can benefit RL by increasing policy entropy (Fig.3). This added entropy enhances exploration by introducing uncertainty, similar to the effect of parameter noise in RL (Plappert et al., 2017; Pang & Jiang, 2021), and helps models discover better strategies (Cui et al., 2025). Our experiments show that a well-designed noise strategy allows quantized LLMs to exploit this effect, reducing memory overhead while gaining better reward curves. This finding contrasts with results from SFT of LLMs (Dettmers et al., 2023a; Guo et al., 2023), demonstrating that controllable quantization noise in LoRA-based RL enhances exploration and enables quantized frameworks to surpass 16-bit LoRA in both efficiency and performance.

Figure 3: Advancement of Quantization in RL. Quantization noise brings higher initialized entropy, which encourages exploration in LoRA-based RL training, accelerating the increase of reward.

We propose QeRL, a quantization-based RL framework designed to train LLMs on reasoning tasks. As shown in Fig.2, QeRL uses NVFP4 quantization for LLM weights and integrates a Marlin-based (Frantar et al., 2024) approach in both rollout and prefilling stages. This design accelerates rollout and prefilling without sacrificing accuracy, with gradient backpropagation enabled through LoRA layers. To address static quantization noise, we introduce adaptive quantization noise (AQN), which injects channel-wise random noise during training and adjusts exploration noise dynamically using an exponential schedule. Additionally, we implement a noise-sharing strategy that merges the noise vector into the layer normalization layer, enabling zero-parameter overhead for noise injection. Compared to vanilla LoRA, QeRL achieves faster rollout and better reward growth. For example, as shown in Fig.1, QeRL outperforms QLoRA and vanilla LoRA in rollout and prefilling speeds on the Qwen2.5-7B-Instruct model, achieving a GSM8K score of 90.8—surpassing both 16-bit LoRA and QLoRA while matching full fine-tuning accuracy on MATH 500. QeRL outperforms vanilla LoRA and QLoRA in both training speed and reward performance. Notably, it achieves approximately a $1.8\times$ speedup in end-to-end training, compared to QLoRA. Additionally, QeRL demonstrates the capability to train a 32B model with GRPO on a single H100 80GB GPU.

## 2 PRELIMINARY

**Model Quantization** Integer quantization requires mapping float-point weights distributed within the interval $[\mathbf{W}_{\min}, \mathbf{W}_{\max}]$ to an integer range of $2^N$, where $N$ is the target bit-width. Given a tensor $\mathbf{W} \in \mathbb{R}^{d \times k}$, this process is defined as:

$$\tilde{\mathbf{W}} = \text{Round}(\frac{\mathbf{W}}{s_{\mathbf{w}}}), s_{\mathbf{w}} = \frac{\mathbf{W}_{\max} - \mathbf{W}_{\min}}{q_{max}} \tag{1}$$

where $\tilde{\mathbf{W}}$ represents the quantized weight matrix, $s_{\mathbf{W}}$ is the scaling factor, and $q_{max}$ defines the compressed range. For integer quantization, $q_{max} = 2^N - 1$. In contrast, for the floating-point quantization, such as FP4 format, $q_{max} = 6$, achieved using a 1-bit mantissa and a 2-bit exponent (E2M1). 4-bit NormalFloat (NF4) is a new data type (Dettmers et al., 2023a), designed for normally distributed weights. Recently, the latest Blackwell GPU architecture (NVIDIA, 2024) introduces hardware support for the advanced FP4 format, MXFP4 (Project, 2023) and NVFP4 (NVIDIA, 2024). MXFP4 adopts a shared FP8 (E8M0) scaling factor across parameter blocks of 32 elements, while NVFP4 employs an FP8 (E4M3) scaling factor with smaller parameter blocks of 16 elements, enabling finer-grained scaling adjustments compared to MXFP4. Both formats are seamlessly integrated into NVIDIA's Hopper (NVIDIA, 2023) and Blackwell (NVIDIA, 2024) GPUs.

**Low-rank Adaptation** LoRA (Hu et al., 2022) is motivated by the observation that weight updates in large pre-trained models often lie in a low-dimensional subspace. Instead of directly fine-tuning all parameters, LoRA introduces a low-rank decomposition to model these updates efficiently:

$$\mathbf{W} + \Delta\mathbf{W} = \mathbf{W} + \mathbf{BA} \tag{2}$$

where $\mathbf{B} \in \mathbb{R}^{d \times r}$ and $\mathbf{A} \in \mathbb{R}^{r \times k}$, with the rank $r \ll \min(d, k)$. In this setup, the original weight matrix $W$ is kept frozen, and only the low-rank matrices $\mathbf{A}$ and $\mathbf{B}$ are optimized during training. This formulation drastically reduces the number of trainable parameters and lowers both memory and computational cost, while retaining the expressivity required for domain adaptation. Within self-attention modules, LoRA is generally applied to the attention and feed-forward projection matrices ($\mathbf{W}_q, \mathbf{W}_k, \mathbf{W}_v, \mathbf{W}_o, \mathbf{W}_{gate}, \mathbf{W}_{up}, \mathbf{W}_{down}$), as these layers are the most critical in LLMs. Other related works are discussed in Appendix D.

## 3 METHOD

Our experiments reveal that quantized LLMs can significantly enhance exploration. Applying parameter-efficient fine-tuning (PEFT) to quantized models not only reduces training resource consumption but also outperforms vanilla LoRA in reward growth and evaluation scores (Fig.2). This challenges the conventional view in SFT that quantization degrades training effectiveness(Dettmers et al., 2023a; Guo et al., 2023). Notably, we observe that quantization error functions similarly to random noise in networks (Plappert et al., 2017; Eberhard et al., 2023; Osband et al., 2016), promoting broader exploration of potential actions or tokens in RL by increasing entropy (Fig.3).

### 3.1 TRAINING FRAMEWORK OF QERL

QeRL is based on the mainstream policy optimization algorithms of LLMs, such as GRPO (Shao et al., 2024) and DAPO (Yu et al., 2025).

**Group Relative Policy Optimization** (Shao et al., 2024) is designed based on the Generalized Advantage Estimation (GAE) (Schulman et al., 2015), eliminating the need for a separately trained reward model, as required in Proximal Policy Optimization (PPO) (Engstrom et al., 2019; Schulman et al., 2017). Instead, for a given input query $q$, multiple samples are generated, resulting in a set of candidate outputs $\{o_1, o_2, ..., o_G\}$. These candidates are evaluated using a rule-based reward, and the average reward is used for updates. The optimization objective is defined as follows:

$$\mathcal{J}(\theta) = \mathbb{E}_{q,\{o_i\}}[\frac{1}{G}\sum_{i=1}^{G}\frac{1}{|o_i|}\sum_{t=1}^{|o_i|}(\min(\frac{\pi_\theta(o_{i,t}|q)}{\pi_{\theta_{old}}(o_{i,t}|q)}A_{i,t}, \text{clip}(\frac{\pi_\theta(o_{i,t}|q)}{\pi_{\theta_{old}}(o_{i,t}|q)}, 1-\alpha, 1+\alpha)A_{i,t})$$
$$-\beta\mathbb{D}_{KL}(\pi_\theta||\pi_{ref}))] \quad (3)$$

where $\pi_\theta$ and $\pi_{ref}$ denote the policy model and reference model, respectively, and the clipping range $(1-\alpha, 1+\alpha)$ stabilized the gradient steps of the policy model. KL penalty is used in GRPO to avoid the unexpected large change in updating (Schulman et al., 2017). $A_{i,i}$ is the antagonist of $i^{th}$ completion, shared across all tokens in $o_t$, defined as:

$$A_i = \frac{r_i - \text{mean}(\{r_1, r_2, ..., r_G\})}{\text{std}(\{r_1, r_2, ..., r_G\})} \quad (4)$$

**Dynamic Sampling Policy Optimization** (Yu et al., 2025) suggests higher clipping upper-bond can help avoid entropy collapse. Another improvement in DAPO is to utilize the loss of token-level policy gradients. In DAPO, the KL penalty from Eq.3 is removed to eliminate the upper limit on exploration in RL, thereby encouraging more optional tokens in the rollout process.

### 3.2 QUANTIZATION ENCOURAGES EXPLORATION

To understand how quantization enhances RL, we analyze its effect on the model's sampling behavior. Our central finding is that the noise introduced by quantization serves as an implicit exploration mechanism, similar to explicit noise injection techniques in the parameter and action space (Plappert et al., 2017; Eberhard et al., 2023; Fortunato et al., 2018; Liu et al., 2025b).

**Quantization Improves Sampling Entropy** We study 3 different quantization formats of FP4 (NVFP4, MXFP4, and NF4) on GSM8K (Cobbe et al., 2021). Our empirical study on Qwen2.5-7B-Instruct (Team, 2024) reveals an intriguing finding: when applying PEFT-based RL, models quantized to 4-bit precision consistently outperform their 16-bit counterparts. This advantage is evident across two key metrics: significantly faster reward convergence during training and higher adjusted evaluation scores. As shown in Fig.4, the reward curves of the models exhibit a steeper upward trend compared to 16-bit models, with convergence patterns closely resembling those of full-parameter fine-tuning in both DAPO and GRPO. Also, NVFP4 and MXFP4 both show better reward growth than NF4.

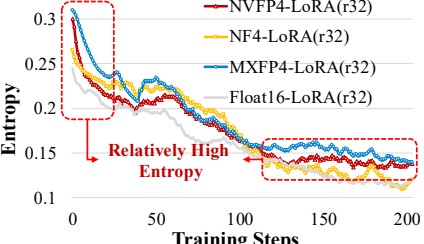

Figure 5: Comparison of RL entropy.

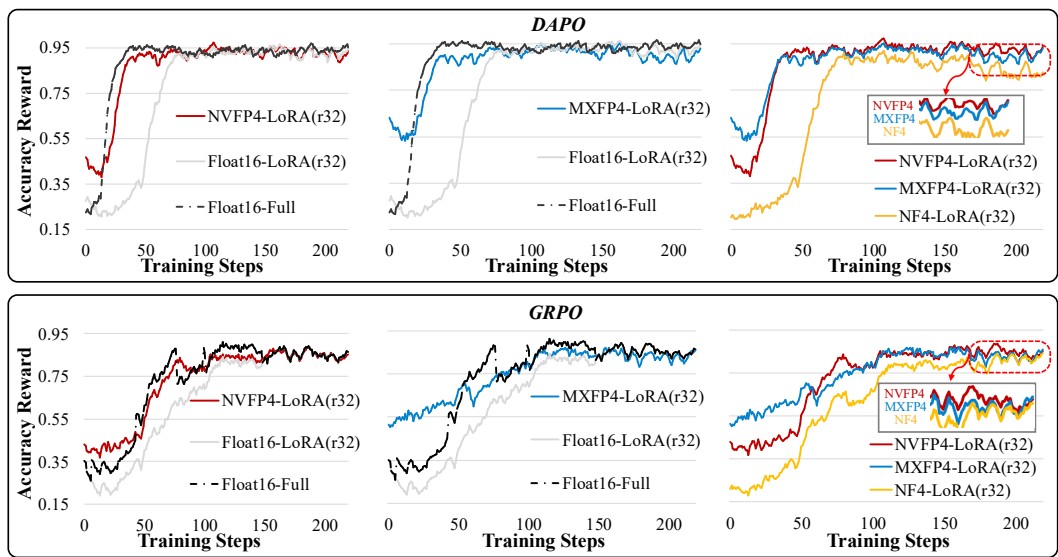

Figure 4: Training reward performance. The upper figures illustrate the training rewards under DAPO, while the lower one is GRPO. Although MXFP4 achieves higher scores in the early stages of training, NVFP4 ultimately converges to better final rewards. LoRA rank is set to 32.

This unexpected performance improvement prompted us to investigate the underlying mechanism. We discover that quantization inherently increases the sampling entropy, $\mathcal{H}(\pi(\cdot|q)) = -\sum_{o_t \in V} \pi(o_t|q) \log \pi(o_t|q)$, where $V$ is the vocabulary) of the policy during deployment (shown in Fig.5). During the forward pass, a quantized model introduces small but systematic errors, which can be modeled as static network noise (Fan et al., 2020). This noise propagates across the network layers, perturbing the final logits before the softmax function is applied. Consequently, the output probability distribution over the vocabulary, denoted as $\pi_\theta(\cdot|q)$, becomes "flatter," with less pronounced peaks. This increase in sampling entropy plays a crucial role in RL by encouraging exploration (Cheng et al., 2025; Eysenbach & Levine, 2021). It mitigates the model's overconfidence in a single "optimal" token and instead assigns more meaningful probabilities to a wider range of plausible next actions (Fig.3). The entropy of other model is provided in Appendix H.

**Quantization Noise** Functionally, this effect resembles exploration in parameters (Eberhard et al., 2023; Plappert et al., 2017), which deliberately injects noise into parameters to drive exploration:

$$(\tilde{\theta} + \theta_{lora}) - (\theta + \theta_{lora}) = Q(\theta) - \theta = \Delta\epsilon \tag{5}$$

where $Q(\theta)$ denotes the de-quantized weight, and $\Delta\epsilon$ is the quantization noise. Such exploratory noise emerges naturally as a computationally "free" byproduct of compressing model representations. This contrasts starkly with SFT, where noise is often detrimental because the objective is to faithfully imitate the true data distribution rather than to discover novel high-reward outputs.

A key limitation of quantization errors is their deterministic nature, which fails to align with the dynamic exploration-exploitation trade-off required in RL. Unlike stochastic noise in traditional RL (Plappert et al., 2017; Osband et al., 2016), which is randomly sampled and independently applied at different training stages, quantization noise remains static throughout the process, lacking the adaptability needed to enhance exploration at critical phases.

## 3.3 ADAPTIVE QUANTIZATION NOISE IN PARAMETER SPACE

To transform static quantization noise into a dynamic exploration mechanism, we introduce an *Adaptive Quantization Noise* (AQN) technique. The core idea is to introduce a small set of structured modulation vectors that slightly perturb the otherwise static quantization noise. In our approach, we utilize an advanced quantization format, NVFP4.

**NVFP4 Quantization** NVFP4 represents weights using a dual-scaling mechanism: a coarse, per-tensor global scaling factor in FP32, $S_{\text{FP32}}$, and a fine-grained tensor of block-wise FP8 (E4M3)

scalers, $\mathbf{S}_{\text{E4M3}}$. The dequantization of a 4-bit $\tilde{\mathbf{W}}$ to the high-precision $\hat{\mathbf{W}}$ follows:

$$\hat{\mathbf{W}} = \text{Dequant}(\tilde{\mathbf{W}}) = S_{\text{FP32}} \cdot (S_{\text{E4M3}} \odot \tilde{\mathbf{W}}) \tag{6}$$

where $\odot$ denotes block-wise scalar multiplication, broadcasting each scaler in $S_{\text{E4M3}}$ to its corresponding block of 4-bit weights in $\tilde{\mathbf{W}}$. The quantization noise of each weight matrix, $\Delta\epsilon = \hat{\mathbf{W}} - \mathbf{W}$, is the difference between this reconstructed tensor and the original full-precision tensor $\mathbf{W}$.

**Adaptive Quantization Noise** We introduce a noise vector to the static quantized weight. Specifically, for each quantized linear layer, we sample a stochastic noise vector, $\mathbf{Z}_{\text{noisy}} \in \mathbb{R}^{1 \times d}$, where $d$ is the input dimension of the layer. This vector is not fixed but is resampled for each forward pass. We define it as: $\mathbf{Z}_{\text{noisy}} = \epsilon, \epsilon \sim \mathcal{N}(0, \sigma^2 I)$, where $\sigma$ is a hyperparameter in different training stage governing the noise scale, and $\epsilon$ is a random vector whose elements are drawn independently from a standard Gaussian distribution (Plappert et al., 2017). Then the additive noise is defined as:

$$\Delta\epsilon' = \mathbf{Z}_{\text{noisy}} + \Delta\epsilon = \mathbf{Z}_{\text{noisy}} + \left(\hat{\mathbf{W}} - \mathbf{W}\right) \tag{7}$$

where $\Delta\epsilon'$ is equivalent to the dynamic noise of each weight matrix. In our setting, we freeze the main branch weight and update the low-rank matrix during RL. The $\mathbf{W}$ and $\hat{\mathbf{W}}$ are consistent values. In the early stages, we leverage the inherent quantization noise to enhance the model's exploration capabilities. As training progresses, $\sigma$ gradually reduces following an exponential decay scheduler:

$$\sigma(k) = \sigma_{\text{start}} \cdot \left(\frac{\sigma_{\text{end}}}{\sigma_{\text{start}}}\right)^{\frac{k-1}{K-1}} \tag{8}$$

where $\sigma_{\text{start}}$ and $\sigma_{\text{end}}$ represent the initial and final noise levels, $k$ is the current stage, and $K$ is the total interval, which are evenly divided in the training steps (more scheduler comparison in Sec.4.2). For instance, our experiments in GSM8K with a total of around 600 training steps, noise is injected at 10 evenly spaced intervals, initialized with quantization noise, then from $\sigma_{\text{start}}$ to $\sigma_{\text{end}}$. This approach aims to balance exploration and exploitation (Fox et al., 2015).

**Noise Merging** While introducing a noise vector enables dynamic control over quantization noise, explicitly creating a separate vector for each quantized layer is not feasible. First, it imposes a burden on parameter efficiency, increasing memory overhead. Moreover, high-precision noise cannot be directly added to quantized weights, as this would break the compatibility of our inference kernel designed for NVFP4 $\times$ BF16 operations. We propose a simple solution that integrates this noise vector directly into the layer normalization parameters of LLM architectures.

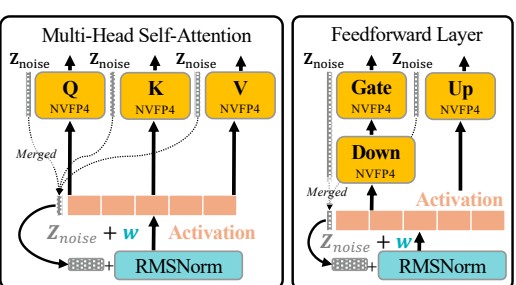

Figure 6: Deployment scheme of adaptive quantization noise in LLMs. $\mathbf{Z}_{noise}$ is integrated in *LayerNorm* (e.g., RMSNorm) of each block in LLMs.

$$\mathbf{X}\left(\mathbf{Z}_{\text{noisy}} + \hat{\mathbf{W}}\right) = \mathbf{X} \cdot \mathbf{Z}_{\text{noisy}} + \mathbf{X} \cdot \hat{\mathbf{W}} \tag{9}$$

By exploiting this equivalency in Eq.9, we subsume the role of $\mathbf{Z}_{\text{noisy}}$ into the learnable weight parameter of the LayerNorm operation (e.g. RMSNorm (Zhang & Sennrich, 2019)) that typically follows the scaling after normalization.

$$\text{RMSNorm}_{noise}(\mathbf{x}) = \mathbf{w}_{noise} \odot \frac{\mathbf{x}}{\sqrt{\frac{1}{N}\sum_{i=1}^{N} x_i^2 + \delta}}, \mathbf{w}_{noise} = \mathbf{Z}_{noise} + \mathbf{w} \tag{10}$$

where $\mathbf{w}$ represents the scaling factor of RMSNorm. In this configuration, channel-wise additive noise $\mathbf{Z}_{\text{noisy}}$ transfers to row-wise multiplicative noise $\frac{\mathbf{Z}_{noise}}{\mathbf{w}} + I$ of weight (proof provided in Appendix G). Multiplicative noise has been shown to be effective in RL (Pang & Jiang, 2021; Zhang et al., 2025a). Due to the higher sensitivity of RL to multiplicative noise, we initialize the noise level with $\sigma_{\text{start}} = 1\text{e-}2$ to ensure stability.

This approach extends adaptive quantization noise to the layer parameters $\mathbf{W}_q$, $\mathbf{W}_k$, $\mathbf{W}_v$, $\mathbf{W}_{\text{gate}}$, and $\mathbf{W}_{\text{up}}$ within each block, as these layers directly interact with normalized activations. To align with LLM architectures (Team, 2024; Grattafiori et al., 2024), $\mathbf{W}_q$, $\mathbf{W}_k$, and $\mathbf{W}_v$ share the same RMSNorm, while $\mathbf{W}_{\text{gate}}$ and $\mathbf{W}_{\text{up}}$ share another (as shown in Fig.6).

(a) Performance of Qwen2.5-3B-Instruct.

| Model | W# | Training | GSM8K |
|---|---|---|---|
| | BF16 | | 61.2 |
| | NF4 | - | $57.5_{-3.7}$ |
| | MXFP4 | - | $59.8_{-1.4}$ |
| | NVFP4 | - | $59.4_{-1.8}$ |
| Qwen2.5-3B | BF16 | Full | $84.4_{+23.2}$ |
| -Instruct | BF16 | LoRA | $76.1_{+14.9}$ |
| | NF4 | LoRA | $76.1_{+14.9}$ |
| | MXFP4 | LoRA | $73.4_{+12.2}$ |
| | NVFP4 | LoRA | $83.3_{+22.2}$ |
| | | +AQN | $83.7_{+22.6}$ |

(b) Performance of Qwen2.5-7B-Instruct.

| Model | W# | Training | GSM8K |
|---|---|---|---|
| | BF16 | - | 76.3 |
| | NF4 | - | $70.5_{-5.8}$ |
| | MXFP4 | - | $71.3_{-5.0}$ |
| | NVFP4 | - | $73.4_{-2.9}$ |
| Qwen2.5-7B | BF16 | Full | $91.2_{+14.9}$ |
| -Instruct | BF16 | LoRA | $88.1_{+11.8}$ |
| | NF4 | LoRA | $85.0_{+8.7}$ |
| | MXFP4 | LoRA | $86.4_{+10.1}$ |
| | NVFP4 | LoRA | $88.5_{+12.2}$ |
| | | +AQN | $90.8_{+13.5}$ |

Table 1: Qwen2.5 Performance on GSM8K. GRPO algorithm is used to train 3B and 7B models on GSM8K dataset, while "Full" denotes the full-parameter training and "W#" represents the bit-width and data format of weight. + and - are compared with original bfloat-16 (BF16) models.



Figure 7: Training reward of 7/14B models.

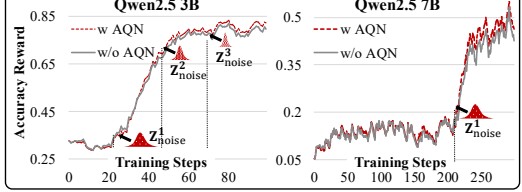

Figure 8: Ablation of AQN on 3/7B model.

## 4 EXPERIMENT

### 4.1 EXPERIMENT SETTINGS

**RL Training** We conducted training experiments using DAPO (Yu et al., 2025) and GRPO (Shao et al., 2024) on two prominent mathematical reasoning datasets: GSM8K (Cobbe et al., 2021) and BigMath (Albalak et al., 2025). GSM8K comprises 7,500 samples with a generation number of 8, while BigMath includes 122,000 samples with a generation number of 16. Both datasets feature problems of medium to high difficulty, spanning levels 3 to 5. For GSM8K, we trained 3B and 7B models, whereas for BigMath, we trained 7B, 14B, and 32B models. Specifically, the 7B and 14B models were trained on problems ranging from levels 3 to 5, while the 32B model was exclusively trained on the more challenging level 4–5 problems. Training checkpoints were evaluated between 500 and 1000 steps. To account for the sensitivity of $Z_{noise}$ perturbation, we set its range from 5e-2 to 5e-4 for dynamic noise estimation. In the main experiments, the LoRA rank is fixed at 32. The speedup tests are performed on a single H100 GPU, while the final evaluated model is trained using 8 H100 GPUs to ensure experimental efficiency on such large-scale data. Detailed hyperparameters and deployment of QeRL are provided in Appendix E and Appendix F.

**Backbone Models** We conduct experiments on Qwen2.5 (Team, 2024) series, using basic without any mathematic data fine-tuning. For weight-only quantization, we applied AWQ (Lin et al., 2024) to MXFP4 and NVFP4 formats. The calibration dataset included 256 sequences, each 2048 tokens long, sampled from OpenThoughts-114k (Guha et al., 2025). Weight-only formats also support inference acceleration on NVIDIA-H100 GPUs with the Marlin kernel (Frantar et al., 2024). For NF4 quantization, we used the default configuration (Dettmers et al., 2023a).

**Evaluation Benchmarks and Metrics** We focus on several widely used mathematical reasoning benchmarks, including GSM8K (Cobbe et al., 2021), MATH500 (Lightman et al., 2023), AIME 2024/2025 (Li et al., 2024), and AMC 23 (Li et al., 2024), for evaluation. During inference, we use a temperature of 0.6, completion length of 4096, and top-p sampling with $p = 0.95$. Each data set is evaluated multiple times, and we report primarily the average accuracy of one sample (Pass@1).

| Model | W# | Training | MATH 500 | AIME 24 | AIME 25 | AMC 23 | Average↑ |
|-------|-----|----------|----------|---------|---------|--------|----------|
| 7B | BF16 | - | 74.8 | 9.2 | 6.6 | 25.0 | 28.9 |
| | NVFP4 | - | $73.7_{-1.3}$ | $8.3_{-0.9}$ | $3.3_{-3.3}$ | $17.5_{-7.5}$ | $25.7_{-3.2}$ |
| | BF16 | Full | $77.4_{+2.6}$ | $16.7_{+7.5}$ | $10.0_{+3.4}$ | $45.0_{+20.0}$ | $37.3_{+8.4}$ |
| | BF16 | LoRA | $77.0_{+2.2}$ | $13.3_{+4.1}$ | $10.0_{+3.4}$ | $42.5_{+17.5}$ | $35.7_{+6.8}$ |
| | NVFP4 | LoRA | $76.8_{+2.0}$ | $13.7_{+4.5}$ | $10.0_{+3.4}$ | $47.5_{+22.5}$ | $37.0_{+8.1}$ |
| | | **+AQN** | $77.4_{+2.6}$ | $15.5_{+6.3}$ | $10.0_{+3.4}$ | $42.5_{+17.5}$ | $36.4_{+7.5}$ |
| 14B | BF16 | - | 78.6 | 11.3 | 9.2 | 45.0 | 36.0 |
| | NVFP4 | - | $76.4_{-2.2}$ | $11.2_{-0.1}$ | $8.3_{-0.9}$ | $40.0_{-5.0}$ | $34.0_{-2.0}$ |
| | BF16 | Full | $83.2_{+4.6}$ | $20.0_{+8.7}$ | $15.1_{+5.9}$ | $55.0_{+10.0}$ | $43.3_{+7.3}$ |
| | BF16 | LoRA | $81.0_{+2.4}$ | $14.0_{+3.7}$ | $13.3_{+4.1}$ | $52.5_{+7.5}$ | $40.2_{+4.2}$ |
| | NVFP4 | LoRA | $79.4_{+0.8}$ | $16.7_{+5.4}$ | $13.3_{+4.1}$ | $52.5_{+7.5}$ | $40.5_{+4.5}$ |
| | | **+AQN** | $80.2_{+1.6}$ | $17.5_{+6.2}$ | $12.6_{+3.4}$ | $57.5_{+12.5}$ | $42.0_{+6.0}$ |
| 32B | BF16 | - | 81.4 | 14.0 | 10.8 | 52.5 | 39.7 |
| | NVFP4 | - | $80.6_{-0.8}$ | $11.3_{-2.7}$ | $10.0_{-0.8}$ | $45.0_{-7.5}$ | $36.7_{-3.0}$ |
| | BF16 | Full | $84.0_{+2.6}$ | $20.0_{+6.0}$ | $23.3_{+12.5}$ | $57.5_{+5.0}$ | $46.2_{+6.5}$ |
| | BF16 | LoRA | $83.6_{+2.2}$ | $16.7_{+3.7}$ | $13.3_{+2.5}$ | $55.0_{+2.5}$ | $42.2_{+2.3}$ |
| | NVFP4 | LoRA | $81.6_{+0.2}$ | $16.7_{+3.7}$ | $15.0_{+4.2}$ | $52.5_{+0.0}$ | $41.4_{+1.7}$ |
| | | **+AQN** | $83.3_{+1.9}$ | $16.7_{+3.7}$ | $19.2_{+8.4}$ | $63.3_{+10.8}$ | $45.6_{+5.9}$ |

Table 2: Performance across four benchmarks. DAPO algorithm is used to train Qwen2.5-7/14/32B-Instruction models on BigMath dataset, while "Full" denotes the full-parameter training.

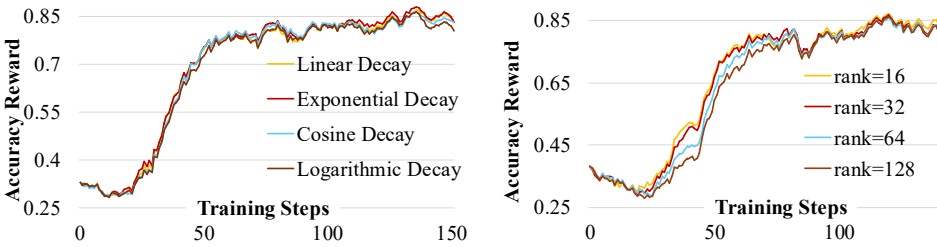

Figure 9: Comparison of noise schedulers.  Figure 10: Ablation of LoRA rank.

## 4.2 EXPERIMENT RESULTS

**Reasoning Performance** As shown in Tab.1, we report the GSM8k training results of the 3B and 7B models using GRPO. While quantized models exhibit performance degradation compared to BF16, applying PEFT with RL to the 3B model demonstrates that NVFP4 combined with AQN achieves a performance of 83.7 from 59.4, surpassing the 76.1 achieved by 16-bit PEFT training and falling only 0.7 points below full-parameter training. Similarly, for the 7B model, our method outperforms 16-bit LoRA by 1.7 points. Furthermore, compared to QLoRA, our approach improves average accuracy by 7.6 and 5.8 points for the 3B and 7B models, respectively. Tab.2 presents the results on the BigMath dataset for the 7B, 14B, and 32B models trained with DAPO. Across all datasets, QeRL consistently matches or exceeds the performance of 16-bit models trained with LoRA. Notably, QeRL trains only about 1% of the parameters required for full-parameter training while using just 40%–50% of the GPU memory of vanilla LoRA. For the 7B model, QeRL improves the average score from 25.7 (quantized) to 36.4, compared to 35.7 with vanilla LoRA. Similar trends are observed in the 14B and 32B models, where QeRL consistently outperforms vanilla LoRA across benchmarks, further supporting the conclusion that quantization enhances RL. Remarkably, on the AMC 23 dataset, the 14B model with QeRL achieves 57.5, exceeding 55.0 of full-parameter training. In Tab. 13, we report both performance and resource consumption for QeRL and FlashRL on GSM8K.

**Reward Visualization** In Sec.3.2, we compare the accuracy rewards of quantized LoRA, vanilla LoRA, and full-parameter training under GRPO and DAPO. Fig.7 presents the accuracy reward curves for the 7B and 14B models on the challenging BigMath dataset. Notably, QeRL achieves a rapid reward increase within 200 steps, while vanilla LoRA requires over 500 steps (Appendix H) to

| Model | Method | W# | Model Size | Training Speedup (Batch Size) | | |
|---|---|---|---|---|---|---|
| | | | | 2 | 4 | 8 |
| Qwen2.5-7B-Instruct | LoRA | BF16 | 15.2 GB | - | - | - |
| | QLoRA | NF4 | 5.7 GB | ×0.8↓ | ×0.8↓ | ×0.7↓ |
| | **QeRL** | **NVFP4** | 5.9 GB | **×1.5↑** | **×1.4↑** | **×1.2↑** |
| Qwen2.5-14B-Instruct | LoRA | BF16 | 29.6 GB | - | - | - |
| | QLoRA | NF4 | 10.2 GB | ×0.9↓ | ×0.7↓ | ×0.7↓ |
| | **QeRL** | **NVFP4** | 10.6 GB | **×1.4↑** | **×1.2↑** | **×1.2↑** |

Table 3: Memory Saving and Speedup of 7B and 14B models. We report the end-to-end speedup in the GRPO process of each training step. Each input has a length of 256 tokens, and each max completion length is 2048. More results of other models are shown in Appendix J.

| Model | Method | Rollout | Reward | Update | Move Model | Total |
|---|---|---|---|---|---|---|
| Qwen2.5-7B-Instruct | BF16 LoRA | 6.28 | 0.05 | 0.30 | 0.57 | 7.20 |
| | QLoRA | 9.48 | 0.06 | 0.74 | 0.15 | 10.43 |
| | QeRL | 4.00 | 0.05 | 0.53 | 0.14 | 4.75 |
| Qwen2.5-14B-Instruct | BF16 LoRA | 10.24 | 0.01 | 0.51 | 0.68 | 11.45 |
| | QLoRA | 12.30 | 0.01 | 1.05 | 0.21 | 13.56 |
| | QeRL | 6.62 | 0.01 | 0.96 | 0.27 | 7.85 |

Table 4: Training Time Breakdown Comparison (s - second). We report the breakdown time in the GRPO process of each training step. Each input has a length of 256 tokens, and each max completion length is 2048.

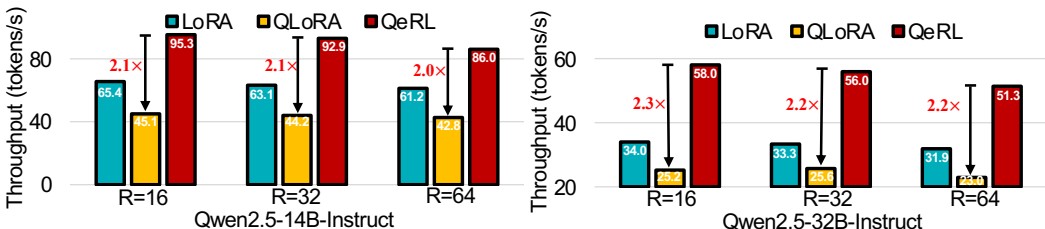

Figure 11: Rollout throughput of 14/32B model. The setting is aligned with Tab. 11 (batch is 1).

show improvement. This finding highlights that the inherent noise introduced by quantized LLMs enhances exploration in LoRA-based RL, enabling faster reward growth and higher reward targets.

**Noise Decay Schedule** Fig.9 compares the performance of different noise decay functions for the 3B model: linear, exponential, cosine, and logarithmic decay. While their performance differences are negligible in the early training stages, exponential decay achieves more stable improvements later by reducing noise to lower levels. The corresponding decay curves are provided in Appendix H.

**Ablation of AQN** Using default quantized noise throughout the training limits the exploration in RL. To address this, we propose the AQN. As shown in Fig.8, when we start with the default quantized noise and periodically inject additional noise in later stages, the reward curve grows more steadily. Notably, when the reward approaches convergence, AQN effectively expands the model's exploration space, enabling further improvements in reward.

**Ablation of LoRA Rank** Fig.10 compares the reward curves of the 3B model during QeRL with different LoRA ranks. Specifically, ranks of 16, 32, 64, and 128 exhibit similar trends and reward growth rates, with rank 16 converging slightly faster, making it a more economical choice.

| (a) Performance on SafeRLHF-QA. | | | |
|---|---|---|---|
| **Model** | **W#** | **Training** | **Acc.** |
| | BF16 | | 78.4 |
| Qwen2.5-3B | BF16 | Full | $87.2_{+8.8}$ |
| -Instruct | BF16 | LoRA | $83.5_{+5.1}$ |
| | **NVFP4** | **QeRL** | $85.3_{+6.9}$ |

| (b) Performance on CommonsenseQA. | | | |
|---|---|---|---|
| **Model** | **W#** | **Training** | **Acc.** |
| | BF16 | | 66.9 |
| Qwen2.5-3B | BF16 | Full | $79.2_{+12.3}$ |
| -Instruct | BF16 | LoRA | $73.5_{+6.6}$ |
| | **NVFP4** | **QeRL** | $73.2_{+6.3}$ |

Table 5: Qwen2.5-3B-Instruct Performance on SafeRLHF-QA and CommonsenseQA. GRPO algorithm is used to train 3B model, while "Acc." denotes the accuracy scores.

## 4.3 MEMORY SAVING AND SPEEDUP

Tab.3 compares the quantized model sizes and end-to-end RL training speedup of these PEFT methods, with all experiments conducted on a single NVIDIA H100-80GB GPU (NVIDIA, 2023). For 7B and 14B models, both QLoRA (NF4) and QeRL (NVFP4, supported by the Marlin kernel (Frantar et al., 2024)) significantly reduce memory usage, shrinking the model sizes to 25%–30% of their 16-bit counterparts. Due to the limitations of NF4 generation speed (Egashira et al., 2024), QLoRA slows to $0.7\times$–$0.8\times$ across different batch sizes. In contrast, QeRL achieves around a $1.2\times$–$1.5\times$ speedup compared to BF16 LoRA in end-to-end RL training while drastically reducing memory usage, and a $1.5\times$–$2.0\times$ speedup compared to QLoRA, benefiting from the generation speed of long reasoning sequences. This efficiency is particularly evident in RL, where the computational demands of long-horizon rollouts emphasize QeRL's advantage. Moreover, detailed breakdown time of QeRL, BF16 LoRA and QLoRA are shown in Tab. 4.

Notably, our speedup measurements are based on the average speed during the first 30 steps, where the output token length is relatively short. In later stages of training, as the model generates longer outputs, the speed advantage of QeRL becomes even more pronounced. Its dual benefits in memory efficiency and training speed make QeRL highly effective for end-to-end RL workflows, especially in scenarios requiring extensive rollouts. Fig.11 shows rollout performance across various LoRA ranks, with QeRL achieving over $2\times$ speedups on 14B and 32B models. On other GPU architectures such as Ampere, one can still apply QeRL using slower dequantization schemes but not NVFP4 * BF16 kernel. Under this setting, the throughput gains are smaller, but we still observe advantages from the reduced memory footprint and improved exploration behavior brought by QeRL. More efficiency comparisons for other models and settings are in Appendix J.

## 4.4 MORE RESULTS ON OTHER RL TASKS

In this section, we extend our experiments beyond math reasoning. The training results of CommonsenseQA (Talmor et al., 2019) (dialogue) and SafeRLHF (Ji et al., 2025) (safety-critical) are reported in Tab.5. We observe that safety-critical tasks often require careful step-by-step analysis to produce safe and accurate responses, so they benefit from reasoning-enhanced RL. In this regime, QeRL consistently outperforms BF16 LoRA, similar to the gains seen on math benchmarks. In contrast, CommonsenseQA relies more on knowledge recall than on complex reasoning, so the benefit from exploration and reasoning-focused RL is less pronounced.

## 5 CONCLUSION

This paper presents QeRL, an efficient training framework for RL on LLMs, which integrates NVFP4 precision quantization with LoRA fine-tuning. The framework is based on the novel observation that quantization can enhance exploration in LoRA-based RL, contrary to findings in SFT. Quantized LLMs not only surpass vanilla 16-bit LoRA training but also approach full-parameter fine-tuning performance. To address the static nature of quantization noise, we introduce an AQN mechanism, which dynamically adjusts noise during training to enhance RL stability. Extensive experiments show that QeRL significantly improves accuracy across models of various sizes compared to both 16-bit LoRA and QLoRA. Additionally, with NVFP4 kernel support, QeRL achieves a round a $1.5\times$ speedup in end-to-end RL training while drastically reducing memory usage.

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

APPENDIX

## A    ETHICS STATEMENT

This work exclusively leverages publicly available open-source datasets that have been previously established and validated in academic research. No new text, video, or audio materials are generated or incorporated as part of this study. The datasets utilized are strictly intended for research purposes and are not employed for any commercial applications.

## B    REPRODUCIBILITY STATEMENT

To ensure the research community can replicate our findings, this project will be released as open-source software. The methodology is described in detail in Sec.3, while Sec.4.1 and Appendix E outline the complete training protocols and implementation details, including all hyperparameter settings.

## C    USE OF LARGE LANGUAGE MODELS

During the preparation of this manuscript, we utilized large language models—GPT-5 (OpenAI, 2025)—exclusively to refine the language, focusing on improving grammar, flow, and tone at the sentence and paragraph levels. These tools were not employed to generate ideas, design experiments, or draw conclusions. All technical content, methodologies, and interpretations were independently written, thoroughly verified, and approved by the authors. To minimize the risk of factual inaccuracies or citation errors, every model-edited sentence underwent human review, and all references were carefully cross-checked with their primary sources. The authors accept full responsibility for ensuring the accuracy and integrity of this manuscript.

## D    RELATED WORK

**Reinforcement Learning for LLMs**    Recent efforts have focused on enhancing reasoning in LLMs using RL (Min et al., 2024; Chu et al., 2025). DeepSeekMath (Shao et al., 2024) improves mathematical reasoning by continuing pre-training on math-intensive data and introducing Group Relative Policy Optimization (GRPO) (Shao et al., 2024). Building on this, DeepSeek-R1 (DeepSeek-AI, 2025) demonstrates that RL alone can drive strong reasoning, achieving performance comparable to proprietary models with large-scale training. Complementary system-level contributions, such as DAPO (Yu et al., 2025), offer an open-source RL framework with a decoupled optimization strategy, achieving competitive results through a simplified training pipeline. GSPO (Zheng et al., 2025) stabilizes RL training and reduces variance through sequence-level optimization, proving effective in large-scale mixture-of-experts models. HybridFlow (Sheng et al., 2025) introduces a flexible RLHF framework with hybrid control flow and a 3D-HybridEngine. Together, these works demonstrate significant progress in advancing LLM reasoning with RL.

**Quantization for LLMs**    Quantization is a key technique for compressing LLMs, improving efficiency by reducing parameter precision. The most common approach, Post-Training Quantization (PTQ) (Dettmers et al., 2022; Frantar et al., 2022; Xiao et al., 2023; Shao et al., 2023; Lin et al., 2024; Chen et al., 2024a; Yuan et al., 2023), transforms pre-trained models cost-effectively without retraining. Vector quantization (Liu et al., 2024b; Tseng et al., 2024; Zhao et al., 2024) is also an effective type to store low-precision data, but still facing the speed problem in inference. Recent work has pushed quantization to ultra-low bit-widths while maintaining performance (Huang et al., 2024c; Dettmers et al., 2023b; Shang et al., 2023; Huang et al., 2024b; Liao & Monz, 2024; Tseng et al., 2024; Huang et al., 2024a; 2026), including advancements in Quantization Aware Training (QAT) to improve robustness (Liu et al., 2023; Chen et al., 2024b). Additionally, novel precision formats like NF4 (Dettmers et al., 2023a), FP4 (Tseng et al., 2025; Chmiel et al., 2025), and MXFP4 (Chmiel et al., 2025) enable accurate weight representation, achieving high compression with minimal or improved accuracy loss. NVFP4 (NVIDIA, 2024) is a groundbreaking 4-bit floating-point format introduced with NVIDIA's Blackwell GPU architecture. This format expands on the idea of compact,

low-bit "micro" floating-point representations, offering developers enhanced versatility by adding another flexible option for their projects (Zhang et al., 2025b; Castro et al., 2025; Lee et al., 2024).

**Efficient Fine-tuning**   Efficient fine-tuning is pivotal for adapting LLMs with minimal computational cost. LoRA (Hu et al., 2022) pioneered this approach by adding low-rank adapters to frozen weight matrices. DoRA (Liu et al., 2024a) improved upon this by decomposing weight updates into directional and magnitude components, addressing low-rank constraints and enhancing stability. QLoRA (Dettmers et al., 2023a) integrated LoRA with 4-bit quantization to further reduce resource usage, while LongLoRA (Chen et al., 2024c) introduced fine-tuning methods for long-context processing. Tina (Wang et al., 2025) demonstrated that compact models could gain reasoning ability through RL with LoRA. Beyond the LoRA family (Hu et al., 2022), other efficient fine-tuning techniques include prompt tuning, prefix tuning, IA3, BitFit, Fisher-masked tuning, and input-tuning (Lester et al., 2021; Li & Liang, 2021; Liu et al., 2022; Zaken et al., 2022; Sung et al., 2021; An et al., 2022; Guo et al., 2023). These advancements underscore the importance of efficient fine-tuning for practical LLM adaptation.

## E   EXPERIMENT HYPERPARAMETERS

**Training Data and Reward Function**   We trained the Qwen2.5-3B-Instruct, Qwen2.5-7B-Instruct, Qwen2.5-14B-Instruct, and Qwen2.5-32B-Instruct models, which are widely used for evaluating reasoning capabilities. Unlike other studies that rely on math-specialized models, we aim to evaluate training performance starting from general-purpose base models. Additionally, QeRL can be smoothly transferred to other model families, such as the Qwen3 series. For the GSM8K dataset, we primarily trained the Qwen2.5-3B-Instruct and Qwen2.5-7B-Instruct models using GRPO, while for the BigMath dataset, we focused on training the Qwen2.5-7B-Instruct, Qwen2.5-14B-Instruct, and Qwen2.5-32B-Instruct models using DAPO. Specifically, for the 7B and 14B models, we selected data with medium to high difficulty levels (grades 3–5), and for the 32B model, we used high-difficulty data (grades 4–5). For problem prompts, we append the suffix `Solve the following math problem step by step.  The reasoning process and direct answer are enclosed within <think> </think> and <answer> </answer> tags, respectively, i.e., <think> reasoning process here </think> <answer> answer here </answer>:  <think> ... </think> <answer> ...  </answer>`.

**RL Training Configuration**   For both GRPO and DAPO, we use the hyperparameters in Tab.6, without using entropy or KL losses. For 4-bit training, the learning rate is set to $1e^{-5}$. However, due to the fragile of the BF16 model with LoRA, the learning rate can not be larger than $5e^{-6}$, or it will collapse in the late training stage.

| Hyperparameter | Value |
|---|---|
| Optimizer | AdamW-8bit |
| Policy learning rate | $1e^{-5}$ (QeRL, QLoRA) / $5e^{-6}$ (LoRA) |
| Training batch size | 128 |
| Samples per prompt | 8 (GSM8K) / 16 (BigMath) |
| Policy updates per rollout | 4 (GSM8K, off-policy) / 1 (BigMath, on-policy) |
| Max response length | 4096 (GSM8K) / 8192 (BigMath) |
| Rollout temperature | 1.0 |
| Clip range $\epsilon_{\text{low}}, \epsilon_{\text{high}}$ | 0.2, 0.28 |
| Noise range $Z_{\text{start}}, Z_{\text{end}}$ | 1e-2, 5e-4 |

Table 6: Hyperparameters of GRPO and DAPO training

## F   DEPLOYMENT OF QERL

In Algorithm 1, we provide a detailed explanation of how QeRL is deployed within the GRPO framework. During the steps in stage 0, the added noise $\sigma$ is set to 0, where only quantization noise

---

**Algorithm 1** Deploy GRPO with QeRL and Adaptive Quantization Noise

---

**Input** NVFP4 policy model $\pi_{\tilde{\theta}}$; reward function $r_\phi$; task prompts $\mathcal{D}$; hyperparameters; LoRA rank, LoRA alpha; number of stages $K$; $\sigma_{\text{start}}, \sigma_{\text{end}}$;

1: policy model $\pi_\theta \leftarrow \pi_{\tilde{\theta}+\theta_{lora}}$
2: **for** iteration = 1, ..., I **do**
3:      reference model $\pi_{ref} \leftarrow \pi_\theta$
4:      **for** step = 1, ..., M **do**
5:          Divide total steps M into $K$ equal stages: steps per stage $= \lfloor M/K \rfloor$
6:          Determine current stage $k$: $k = \lfloor \frac{\text{step}-1}{\text{steps per stage}} \rfloor$
7:          **Set noise level** $\sigma \leftarrow \begin{cases} 0 & \text{if } k = 0 \\ \sigma_{\text{start}} \cdot \left(\frac{\sigma_{\text{end}}}{\sigma_{\text{start}}}\right)^{\frac{k-1}{K-1}} & \text{otherwise (exponential decay)} \end{cases}$
8:          Sample a batch $\mathcal{D}_b$ from $\mathcal{D}$
9:          Update the old policy model with AQN: $\pi_{\theta_{old}} \leftarrow \pi_\theta + \mathcal{N}(0, \sigma^2)$
10:         Sample $G$ outputs $\{o_i\}_{i=1}^G \sim \pi_{\theta_{old}}(\cdot \mid q)$ for each question $q \in \mathcal{D}_b$
11:         Compute rewards $\{r_i\}_{i=1}^G$ for each sampled output $o_i$ by running $r_\phi$
12:         Compute $\hat{A}_{i,t}$ for the $t$-th token of $o_i$ through group relative advantage estimation.
13:         **for** GRPO iteration = 1, ..., $\mu$ **do**
14:            Update the policy model $\pi_\theta$ by maximizing the GRPO objective (Equation 3)
15:         **end for**
16:      **end for**
17: **end for**
**Output** $\pi_\theta$

---

effects. At stage 1, $\sigma$ is initialized to $\sigma_{\text{start}}$, and by the final stage (K-1) $\sigma$ gradually transitions to $\sigma_{\text{start}}$. This progressive adjustment of noise ensures a structured and controlled exploration process throughout the training stages, balancing stability and exploration effectively.

## G   PROOF OF NOISE SHARING

In this section, we further demonstrate the effectiveness of the noise-sharing operation proposed in Eq.10, detailing the process by which additive noise is transformed into multiplicative noise. With AQN, input of each block follows:

$$\text{RMSNorm}_{noise}(\mathbf{X}) = \left(\frac{\mathbf{Z}_{noise}}{\mathbf{w}} + I\right) \odot \text{RMSNorm}(\mathbf{X}), \tag{11}$$

where $\text{RMSNorm}(\cdot)$ denotes the vanilla RMSNorm operation and $\mathbf{w}$ is the original scaling factor in $\text{RMSNorm}(\cdot)$. The element-wise multiplication ($\odot$) will be auto-broadcast during computing. Then, the operation of the following linear computation is defined as:

$$\left(\left(\frac{\mathbf{Z}_{noise}}{\mathbf{w}} + I\right) \odot \text{RMSNorm}(\mathbf{X})\right) \cdot \hat{\mathbf{W}} = \text{RMSNorm}(\mathbf{X}) \cdot \left(\left(\frac{\mathbf{Z}_{noise}}{\mathbf{w}} + I\right)^\top \odot \hat{\mathbf{W}}\right), \tag{12}$$

Thus, the additive Gaussian noise, when incorporated into the noise-sharing mechanism of Layer-Norm, can be equivalently regarded as multiplicative Gaussian noise (denoted as $\left(\frac{\mathbf{Z}_{noise}}{\mathbf{w}} + I\right)$) and applied row-wise to the weight matrix $\hat{\mathbf{W}}$. Since RMSNorm is only applied to the inputs of each attention block and feed-forward network (FFN) block, this mechanism ensures that the Q, K, and V matrices in the attention block share the same noise, while the down and up layers in the FFN block also share a single, identical noise set. This noise-injection strategy avoids disrupting the multiplication kernels of NVFP4 and BF16 in QeRL or introducing additional matrix multiplication operations.

Both additive and multiplicative noise have been shown to positively contribute to exploration in RL (Plappert et al., 2017; Higuera et al., 2018; Chen et al., 2025b). However, multiplicative noise tends to be more sensitive, especially in deep networks like LLMs. To address this, we initialize the noise standard deviation ($\sigma$) to 1e-2, which is smaller than the typical 1e-1 used in traditional noise-based networks.

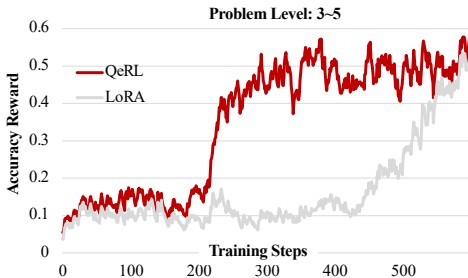

Figure 12: Training reward of 7B model.

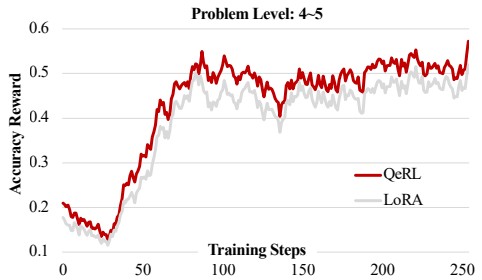

Figure 13: Training reward of 32B model.

## H  ADDITIONAL EXPERIMENTS OF TRAINING

**Training Rewards of Different Model**   Fig.12 and Fig.13 further compare the performance of QeRL and 16-bit LoRA training on complex reasoning datasets. In Fig.12, we present the training rewards of the Qwen2.5-7B-Instruct model on the BigMath dataset with difficulty levels ranging from 3 to 5, as an extension of Fig.7. Leveraging the exploration benefits of QeRL in quantized models, a rapid increase in reward is observed after approximately 200 steps, whereas 16-bit LoRA requires over 500 steps to achieve a similar rise. Meanwhile, as shown in Fig.13, we trained the Qwen2.5-32B-Instruct model on the highest difficulty data (levels 4–5). Although the difference in reward growth between QeRL and LoRA is less pronounced in the 32B model compared to the smaller 3B, 7B, and 14B models, QeRL still consistently performs better than LoRA.

**More Experiments of Entropy**   As an extension of Fig.5, Fig.14 illustrates the entropy curve of the Qwen2.5-14B-Instruct model at various training steps. Notably, the entropy of QeRL remains consistently higher than that of LoRA throughout the RL process, particularly during the initial steps. This observation highlights the advantage of QeRL in promoting exploration, as higher entropy indicates a broader search of the solution space. The increased exploratory capacity facilitated by quantization appears to enable the model to navigate complex environments more effectively, ultimately supporting improved

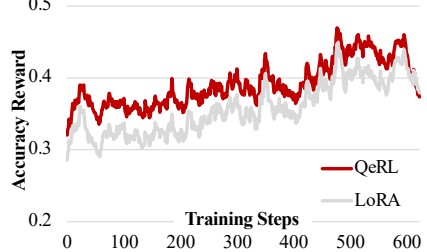

Figure 14: Entropy in RL steps.

optimization. These results further validate the role of quantization in enhancing the exploration-exploitation balance in RL tasks.

**Noise Scheduler**   Fig.15 illustrates the noise scheduler employed in our experiments, showing four distinct decay strategies: linear, exponential, cosine, and logarithmic. The scheduler adjusts the noise level in 10 stages to guide the training process. The linear decay method reduces noise uniformly across stages, ensuring a consistent rate of change. The exponential decay rapidly decreases the noise at the beginning and uses smaller noise scales in later stages, which we found effective for achieving stable and higher rewards in later stages of training. The cosine decay follows a smooth oscillatory pattern, gradually reducing noise with a cosine curve, whereas the logarithmic decay decreases noise

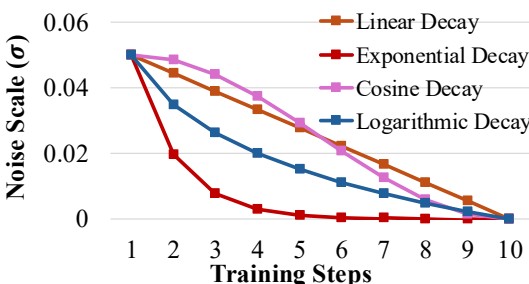

Figure 15: Noise curve of different schedulers.

sharply in early stages and stabilizes in later ones. Among these, we chose the exponential decay strategy due to its ability to maintain smaller noise scales during the later stages, resulting in a more stable and higher reward curve. This flexibility in controlling noise levels plays a critical role in balancing exploration and convergence during training.

We use start noise levels in the range [1e-1, 1e-2] and end noise levels in the range [1e-3, 5e-4]. These are typical scales in traditional RL, where they provide stable exploration without strongly

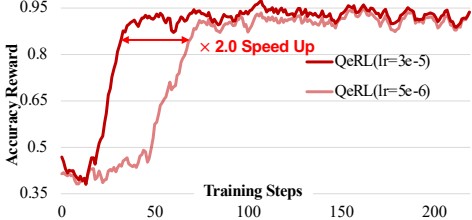

Figure 16: Ablation of learning rate in QeRL (Qwen2.5-7B-Instruct).

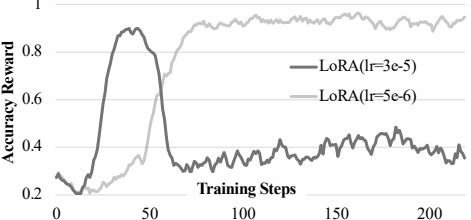

Figure 17: Ablation of learning rate in LoRA (Qwen2.5-7B-Instruct).

| Model | Setting | GSM8K | MATH500 |
|---|---|---|---|
| | - | 76.3 | 74.8 |
| | Full | 91.2 | 77.4 |
| | + AQN | 91.4 | 77.2 |
| | BF16 LoRA | 88.1 | 77.0 |
| | + AQN | 89.7 | 77.8 |
| Qwen2.5-7B-Instruct | MXFP4 LoRA | 86.4 | - |
| | + AQN | 88.1 | - |
| | NF4 LoRA | 85.0 | - |
| | + AQN | 85.7 | - |
| | NVFP4 LoRA | 88.5 | 76.8 |
| | + AQN | 90.8 | 77.4 |

Table 7: More Ablation of ANQ.

| Model | start | end | GSM8K |
|---|---|---|---|
| | 1e-1 | 1e-3 | 82.5 |
| Qwen2.5-3B-Instruct | 1e-2 | 1e-3 | 83.7 |
| | 1e-1 | 5e-4 | 83.1 |
| | 1e-2 | 5e-4 | 83.7 |

Table 8: Comparison of Different Noise Scale.

disrupting the policy. We observe similar behavior in LLM RL: within this range, training remains stable and performance does not change sharply. To make this more quantitative, we added a small sensitivity study on GSM8K with Qwen2.5-3B-Instruct in Tab. 8. These results show that AQN is reasonably robust within this band, with all settings giving stable training and similar performance. We also notice that too large noise (above 1e-1) can start to hurt performance, so we keep the start noise at or below 1e-1 in all our main experiments.

Across full-parameter tuning and different quantization formats (NF4, MXFP4, BF16) with LoRA, adding AQN consistently gives modest gains, which is consistent with parameter-noise based exploration in (Fortunato et al., 2018; Plappert et al., 2017). We also observe that the relative improvement is larger in the LoRA setting. Our interpretation is that full-parameter RL can already fit the data well, leaving less headroom for additional exploration, whereas in low-rank (LoRA) updates, AQN-induced exploration provides a stronger benefit.

## I  ADDITIONAL ABLATION STUDY

**Ablation of Learning Rate** We examine the impact of learning rate variations on the performance of quantized models compared to 16-bit models. As illustrated in Fig.16 and Fig.17, with a relatively small learning rate of 5e-6, QeRL marginally outperforms LoRA, achieving a reward close to 0.95. When the learning rate is increased to 3e-5, the larger update magnitude in the adapter results in faster reward growth and quicker model convergence. However, in 16-bit models, the excessive update magnitude leads to instability, often causing the training process to collapse. In contrast,

| Method | W# | Model Size | BS# | Throughput (Tokens/s) | | E2E RL Speedup | |
|--------|-----|-----------|-----|----------------|---------|---------|---------|
| | | | | Rollout Phase | Speedup | w/o GC | w/ GC |
| LoRA | BF16 | 6.2 GB | 2 | 151.2 | - | - | - |
| **QeRL** | **NVFP4** | 2.8 GB | 2 | **157.0** | ×1.0 | ×1.1 | ×1.0 |
| LoRA | BF16 | 6.2 GB | 8 | 2226.3 | - | - | - |
| **QeRL** | **NVFP4** | 2.8 GB | 8 | **2271.4** | ×1.0 | ×1.1 | ×1.1 |

Table 9: Memory Saving and Speedup of Qwen2.5-3B-Instruct Model. The table reports the throughput (tokens/s) for the rollout phase under two batch size settings (2 and 8). Each input has a length of 256 tokens, and each max completion length is 2048. "W#" denotes the data format, "BS#" is the number of batch size, and "E2E" denotes the end-to-end speed of GRPO training. "GC" denotes gradient checkpointing.

| Method | W# | Model Size | BS# | Throughput (Tokens/s) | | E2E RL Speedup | |
|--------|-----|-----------|-----|----------------|---------|---------|---------|
| | | | | Rollout Phase | Speedup | w/o GC | w/ GC |
| LoRA | BF16 | 15.2 GB | 2 | 115.4 | - | - | - |
| **QeRL** | **NVFP4** | 5.9 GB | 2 | **151.6** | ×1.3 ↑ | ×1.2 ↑ | ×1.2 ↑ |
| LoRA | BF16 | 15.2 GB | 8 | 1641.1 | - | - | - |
| **QeRL** | **NVFP4** | 5.9 GB | 8 | **2091.8** | ×1.3 ↑ | ×1.1 ↑ | ×1.1 ↑ |

Table 10: Memory Saving and Speedup of Qwen2.5-7B-Instruct Model. The table reports the throughput (tokens/s) for the rollout phase under two batch size settings (2 and 8). Each input has a length of 256 tokens, and each max completion length is 2048. "W#" denotes the data format, "BS#" is the number of batch size, and "E2E" denotes the end-to-end speed of GRPO training. "GC" denotes gradient checkpointing.

| Method | W# | Model Size | BS# | Throughput (Tokens/s) | | E2E RL Speedup | |
|--------|-----|-----------|-----|----------------|---------|---------|---------|
| | | | | Rollout Phase | Speedup | w/o GC | w/ GC |
| LoRA | BF16 | 29.6 GB | 2 | 65.4 | - | - | - |
| **QeRL** | **NVFP4** | 10.6 GB | 2 | **95.3** | ×1.3 ↑ | ×1.4 ↑ | ×1.4 ↑ |
| LoRA | BF16 | 29.6 GB | 8 | 737.2 | - | OOM | - |
| **QeRL** | **NVFP4** | 10.6 GB | 8 | **1091.1** | ×1.5 ↑ | OOM | ×1.3 ↑ |

Table 11: Memory Saving and Speedup of Qwen2.5-14B-Instruct Model. The table reports the throughput (tokens/s) for the rollout phase under two batch size settings (2 and 8). Each input has a length of 256 tokens, and each max completion length is 2048. "W#" denotes the data format, "BS#" is the number of batch size, and "E2E" denotes the end-to-end speed of GRPO training. "GC" denotes gradient checkpointing.

QeRL demonstrates remarkable robustness to larger learning rates due to the presence of NVFP4 quantization noise, which helps stabilize updates. This robustness enables QeRL to maintain stable training even under high learning rates, achieving a reward growth rate nearly twice as fast as the 16-bit model. These results underscore QeRL's superior adaptability and efficiency, particularly in challenging training scenarios with high learning rates.

## J   MORE EFFICIENCY EXPERIMENTS

Tab.9, Tab.10, Tab.11, and Tab.12 provide additional speed benchmarks for the Qwen2.5-3B-Instruct, Qwen2.5-7B-Instruct, Qwen2.5-14B-Instruct, and Qwen2.5-32B-Instruct models, evaluated under batch sizes of 2 and 8. For the 3B and 7B models, we did not enable memory-efficient techniques such as gradient checkpointing (Chen et al., 2016) or Liger loss (Hsu et al., 2025) in

| Method | W# | Model Size | BS# | Throughput (Tokens/s) | | E2E RL Speedup | |
|--------|-----|-----------|-----|---------------|---------|---------|---------|
| | | | | Rollout Phase | Speedup | w/o GC | w/ GC |
| LoRA | BF16 | 62.3 GB | 2 | 34.0 | - | OOM | OOM |
| **QeRL** | **NVFP4** | 20.7 GB | 2 | **60.0** | **×1.8** | OOM | 10.6 s/step |
| LoRA | BF16 | 62.3 GB | 8 | 344.3 | - | OOM | OOM |
| **QeRL** | **NVFP4** | 20.7 GB | 8 | **688.2** | **×2.0** | OOM | 12.2 s/step |

Table 12: Memory Saving and Speedup of Qwen2.5-32B-Instruct Model. The table reports the throughput (tokens/s) for the rollout phase under two batch size settings (2 and 8). Each input has a length of 256 tokens, and each max completion length is 2048. "W#" denotes the data format, "BS#" is the number of batch size, and "E2E" denotes the end-to-end speed of GRPO training. "GC" denotes gradient checkpointing.

| Model | Method | Rollout W# | Memory | GSM8K |
|-------|--------|-----------|--------|-------|
| | Baseline | - | - | 76.3 |
| Qwen2.5-7B-Instruct | FlashRL (w TIS) | FP8 | > 70 GB | 89.2 |
| | **QeRL** | **NVFP4** | > 15 GB | **90.8** |

Table 13: Comparison of QeRL and FlashRL.

| Model | BF16 (Tokens/s) | | | Model | NVFP4 (Tokens/s) | | |
|-------|---------|---------|---------|-------|---------|---------|---------|
| | Rank 16 | Rank 32 | Rank 64 | | Rank 16 | Rank 32 | Rank 64 |
| 3B | 151.2 | 148.8 | 138.6 | 3B | 157.0 | 153.1 | 140.0 |
| 7B | 115.4 | 113.2 | 108.3 | 7B | 151.6 | 149.9 | 137.7 |
| 14B | 65.4 | 63.1 | 61.2 | 14B | 95.3 | 92.9 | 86.0 |
| 32B | 34.0 | 33.3 | 31.9 | 32B | 58.0 | 56.0 | 51.3 |

Table 14: Throughput under different LoRA ranks in the rollout stage. We test the tokens/s for each model in the vLLM engine, and the setting is aligned with Tab.11. We set the batch size as 1.

order to maximize training speed. However, due to the substantial size of the 14B and 32B models and the computational overhead introduced by importance sampling with gradients during RL training, we employ gradient checkpoint to accelerate computation. For training on GPUs with smaller memory capacity, enabling gradient checkpointing is recommended to reduce memory usage, although this may come at the cost of slower overall training speed. During the rollout phase, the precision of NVFP4, optimized by the Marlin kernel (Frantar et al., 2024), demonstrates a significant acceleration, achieving speeds of 1.0 to 2.0×. In particular, performance gains become more pronounced as model size increases, with the 32B model achieving up to a 2.0× speedup. This indicates that NVFP4's advantages are particularly impactful for large-scale models, where computational demands are higher.

In end-to-end RL efficiency evaluation, we report the per-step latency of GRPO training, defined as the wall clock time to complete an optimization step including rollout generation, log-probability computation, and parameter updates. We benchmark with rollout batch sizes of 2 and 8 while fixing the maximum input length to 256 tokens and the maximum completion length to 2,048 tokens. For fairness, we match the vLLM memory budget between BF16 and NVFP4 variants by setting the same gpu memory utilization in the engine: 0.20 for Qwen2.5-3B-Instruct, 0.30 for 7B, 0.45 for 14B, and 0.40 for 32B (the latter to enable single-GPU training). Under these controlled settings, the E2E latency reductions mirror the rollout phase acceleration and become more pronounced as the model size grows, with the largest gains observed on Qwen2.5-14B-Instruct.

Additionally, Tab.14 provides a comparison of inference speeds between 16-bit and NVFP4 main models across various LoRA ranks. NVFP4 consistently outperforms 16-bit models in terms of speed at all adapter ranks, showcasing its ability to maintain efficiency across diverse configurations. However, as the rank increases, both NVFP4 and BF16 experience a gradual decline in rollout

speed within the vLLM engine, likely due to the increased computational overhead associated with higher ranks. Despite this, NVFP4 continues to demonstrate superior performance, highlighting its robustness and adaptability for both small-scale and large-scale setups. These findings underscore NVFP4's potential to optimize inference efficiency, particularly when combined with advanced kernels and varying adapter configurations.

## K    LIMITATION ANALYSIS

We have demonstrated that our method, QeRL, achieves superior performance in RL training for LLMs compared to 16-bit vanilla LoRA training. Additionally, QeRL matches the accuracy of 16-bit full-parameter reinforcement fine-tuning while delivering over $2\times$ training speedup relative to both vanilla LoRA and QLoRA. However, since RL for LLMs inherently demands significantly greater computational resources than SFT, our experiments, conducted on model sizes ranging from 3B to 32B, do not yet establish whether QeRL can maintain the same level of performance for models exceeding 70B parameters, leaving that investigation for future work. Another limitation is that RL training often requires tens or even hundreds of hours, and while we have provided comprehensive evaluations on reasoning benchmarks such as GSM8K, MATH 500, AIME 24, AIME 25, and AMC 23, we did not extend our evaluations to other benchmarks or data types, such as code, or to general-purpose language tasks unrelated to reasoning. Nevertheless, our technique can be seamlessly adapted to richer and more diverse training datasets. We encourage the community to explore and apply this method to a broader range of tasks in future research.

