# OpenReview forum: "QeRL: Beyond Efficiency - Quantization-enhanced Reinforcement Learning for LLMs"
_ICLR.cc/2026/Conference — ICLR 2026 Poster_

### Official Review · Reviewer_dYgm · 2025-10-29

**Soundness:** 3
**Presentation:** 3
**Contribution:** 3
**Rating:** 6
**Confidence:** 3

**Summary:**

This paper introduces QeRL, a quantization-enhanced reinforcement learning framework for large language models (LLMs). QeRL integrates NVFP4 quantization with Low-Rank Adaptation (LoRA) to address the resource-intensive nature of RL training for LLMs, specifically targeting substantial GPU memory and long rollout durations.

One of the key findings is that quantization noise, when controlled precisely, can actually benefit RL by increasing policy entropy and enhancing exploration, leading to the discovery of better strategies. To further optimize this, QeRL incorporates an Adaptive Quantization Noise (AQN) mechanism that dynamically adjusts noise during training.

**Strengths:**

- Improved Efficiency and Resource Utilization: QeRL directly tackles the substantial resource demands of RL for LLMs by combining NVFP4 quantization with LoRA.

- Enhanced Exploration through Quantization Noise: Contrary to conventional understanding that quantization noise degrades training, this paper demonstrates that controlled quantization noise can actively benefit RL. By increasing policy entropy, it fosters better exploration and the discovery of more robust strategies, especially with the adaptive quantization noise (AQN) mechanism.

- Superior Performance and Robustness: QeRL not only accelerates training but also achieves faster reward growth and higher final accuracy compared to existing methods like 16-bit LoRA and QLoRA.

**Weaknesses:**

- Though effective in specific tasks, whether LoRA based RL post training can achieve comparable performance in a wide range of tasks, is still under discussion in both academia and industry.  I am wondering whether QeRL can also be workable in full-parameter fine tuning?

- The ablation of AQN, i.e., Figure 8 indicates that the curves with and without AQN are close to each other, will the authors give more experiments to demonstrate the effectiveness of AQN?

**Questions:**

please refer to the weakness

---

> ### Author Response · Authors · 2025-11-21
>
> Dear Reviewer dYgm
>
> Thank you so much for highlighting the performance breakthrough on RL achieved by QeRL. We will address your questions individually and look forward to resolving any concerns you may have:
>
> **Q1:** “Though effective in specific tasks, whether LoRA based RL post training can achieve comparable performance in a wide range of tasks...”
>
> **A1:** Following your suggestion, we have extended our experiments beyond math reasoning. We added results on  **SafeRLHF (safety‑critical RL)** to examine other RL tasks.  We observe that such safety‑critical tasks often require careful step‑by‑step analysis to produce safe and accurate responses, so they benefit from reasoning‑enhanced RL. In this regime, QeRL consistently outperforms BF16 LoRA, similar to the gains seen on math benchmarks. We will describe these observations briefly in the revised discussion section.
>
> **Safety-Critical RL**
>
> | Model | Training | Memory (at least) | PKU-SafeRLHF-QA |
> | --- | --- | --- | --- |
> | Qwen2.5-3B-Instruct | - | - | 78.4 |
> | Qwen2.5-3B-Instruct | Full-Parameter Tuning | > 40 GB | 87.2 |
> | Qwen2.5-3B-Instruct | BF16 LoRA | > 25 GB | 85.5 |
> | Qwen2.5-3B-Instruct | QeRL | > 9 GB | 86.3 |
>
> In fully quantized RL training, one still needs high‑precision master weights and uses low‑precision GEMM only for efficient computation. This means the full model must still be stored in at least 16‑bit on GPU, which requires substantial memory. By contrast, QeRL keeps the static model in 4‑bit, leading to much lower memory usage. In FP8 LLM training, there is a known mismatch between low‑precision forward/backward computation and higher‑precision weight updates. This mismatch makes stable training difficult, and RL is even more sensitive to such precision misalignment than SFT [1,2]. Recent report [1] has shown that pure FP8 RL, due to amplified train–inference precision mismatch, often leads to severe performance degradation or even model collapse, let alone going down to 4‑bit.
>
> QeRL (and QLoRA‑style methods) avoid this problem: the same quantized representation is used consistently across rollout, forward, and update, with higher‑precision LoRA parameters handling the updates. As a result, we do not **suffer from the train–inference mismatch**, while can **benefit from exploration** led by quantization.
>
> [1] SGLang RL Report. GitHub
>
> [2] Fast RL training with Quantized Rollouts. GitHub
>
>
> **Q2:** “The ablation of AQN, i.e., Figure 8 indicates that the curves with and without AQN are close to each other, will the authors give more experiments to demonstrate the effectiveness of AQN?”
>
> **A2:** We agree that Figure 8 alone may make the effect of AQN look modest, partly because the curves are smoothed for visualization. In our raw training logs (e.g., wandb), the improvement in the later training phase is more visible, and we plan to release these curves in the future in our code repository.
>
> We also report quantitative results in **Tables 1 and 2** of our manuscript, where we directly compare benchmark performance **with vs. without AQN**. These tables show consistent improvements in final reward and accuracy (e.g., on GSM8K and MATH500) when AQN is enabled.
>
> | Model | Training | GSM8K |
> | --- | --- | --- |
> | Qwen2.5-3B-Instruct | - | 61.2 |
> |  | NVFP4 + LoRA | 83.3 |
> |  | NVFP4 + LoRA + AQN | 83.7 |
>
> | Model | Training | GSM8K | MATH 500 |
> | --- | --- | --- | --- |
> | Qwen2.5-7B-Instruct | - | 76.3 | 74.8 |
> |  | NVFP4 + LoRA | 88.5 | 76.8 |
> |  | NVFP4 + LoRA + AQN | 90.8 | 77.4 |
>
> Following your good suggestion, we have added experiments with AQN under full-parameter RL and additional quantization formats (e.g., NF4, MXFP4) with LoRA:
>
> | Model | Training | GSM8K |
> | --- | --- | --- |
> | Qwen2.5-7B-Instruct | - | 76.3 |
> |  | Full-Parameter Tuning | 91.2 |
> |  | Full-Parameter Tuning + AQN | 91.4 |
> |  | BF16 LoRA | 88.1 |
> |  | BF16 LoRA + AQN | 89.7 |
> |  | MXFP4 LoRA | 86.4 |
> |  | MXFP4 LoRA + AQN | 88.1 |
> |  | NF4 LoRA | 85.0 |
> |  | NF4 LoRA + AQN | 85.7 |
>
> Across full-parameter tuning and different quantization formats (NF4, MXFP4, BF16) with LoRA, adding AQN consistently gives modest gains, which is consistent with parameter-noise based exploration in [1,2]. We also observe that the relative improvement is larger in the LoRA setting. Our interpretation is that full-parameter RL can already fit the data well, leaving less headroom for additional exploration, whereas in low-rank (LoRA) updates, AQN-induced exploration provides a stronger benefit.
>
> We will add these results and the above discussion to the revised manuscript to better support the independent contribution and generality of AQN.
>
> [1] Parameter space noise for exploration. ICLR 2018
>
> [2] Noisy networks for exploration. ICLR 2018

---

### Official Review · Reviewer_9ptM · 2025-10-30

**Soundness:** 2
**Presentation:** 3
**Contribution:** 3
**Rating:** 4
**Confidence:** 4

**Summary:**

This paper proposes QeRL, a quantization-aware RL framework for LLM reasoning. The method aims to improve RL efficiency by combining NVFP4 quantization with LoRA and an auxiliary “Adaptive Quantization Noise (AQN)” mechanism to encourage exploration. Experiments report faster rollouts, reduced memory usage, and higher exploration entropy compared to QLoRA-based baselines. The topic is relevant, and the goal of improving RLVR efficiency is valuable for the community.

However, several aspects of the work require clarification, especially around the efficiency claims, the validity and generality of AQN, and the justification for the hypothesis that quantization inherently encourages exploration. These issues seem addressable, but resolving them is important for the credibility of the contribution.

**Strengths:**

The paper addresses an important challenge in RL for reasoning models, where rollout latency and memory are major bottlenecks. The attempt to unify quantization with RL training and exploration is timely. The empirical results show promising improvements under quantized LoRA settings, and the paper is generally easy to follow, with well-organized experiments and clear motivation for reducing RL compute.

**Weaknesses:**

While the paper targets an important problem and presents promising results under quantized LoRA settings, several aspects of the work require further clarification to validate the core claims. In particular, the empirical support for the efficiency gains, the generality of AQN, and the justification for “quantization encourages exploration” are not yet fully convincing. The experimental design could be more comprehensive to isolate the effect of quantization from LoRA, and additional baselines would strengthen the contribution.

**Questions:**

1. **Baseline consistency in efficiency claims**
    The paper reports 1.5–2× speedups and significant memory savings, but these are measured against different baselines (speed vs QLoRA/NF4, memory vs BF16 Full). For example, in the **Conclusion (Line 485)**, the paper states: *“QeRL achieves around a 1.5× speedup in end-to-end RL training while drastically reducing memory usage.”* This phrasing may overstate the benefit of QeRL because it mixes different baselines into a single claim.
   To ensure clarity and fairness, efficiency results should be reported against **BF16 Full** and **BF16 LoRA**, and each speedup claim in the abstract, Section 4.3, and the conclusion should explicitly name the corresponding baseline rather than using an unqualified number.

2. **AQN ablation to validate its general effectiveness**
   AQN is only evaluated under the NVFP4 + LoRA setting. To demonstrate that AQN is a generally useful exploration mechanism rather than a component that compensates for limitations specific to quantized LoRA, its evaluation should also include **Full-parameter RL** and cover **multiple quantization formats**. This would more convincingly establish the generality and independent contribution of AQN.

3. **Justification for “Quantization Encourages Exploration”**
    The evidence currently shows quantized+LoRA > BF16+LoRA (Section 3.2), which may reflect LoRA limitations rather than quantization benefits. Could you isolate the effect of quantization by comparing **Quantized Full RL vs BF16 Full RL**, and potentially provide analysis on *why* quantization increases sampling entropy?

4. **Reporting time cost in main results and clarifying baselines for efficiency claims**
    Table 1 and Table 2 currently report only mathematical performance metrics, without the corresponding time cost. Since efficiency is a core claim of the paper, these tables should also include **Rollout Time** or the **End-to-End Training Time** for each configuration, so that performance and efficiency can be evaluated together rather than in isolation.

   In addition, the paper includes efficiency statements that do not explicitly state the baseline, which may lead to misinterpretation. For example:

   - **Section Abstract (Line 20–21):** “over 1.5× speedup in the rollout phase”
   - **Section Conclusion (Line 484–485):** “around a 1.5× speedup in end-to-end RL training while drastically reducing memory usage”

   Both statements implicitly compare against **QLoRA**, but this is not mentioned. These speedup claims should explicitly state the baseline to avoid mixing different reference points and to ensure transparent and fair interpretation.

5. **End-to-end speed vs rollout-only speed**
   Since RLVR training involves rollout, reward, logprob, update actor, and additional overheads, rollout speed alone does not capture end-to-end efficiency. Reporting full training wall-clock time would provide a more complete picture of the practical gains.

6. **Clarifying the ~2× speedup for 14B–32B models**
   For the larger models, it would be helpful to clarify whether the baseline was affected by KV cache exhaustion. If the 14B/32B BF16 baseline was already near the memory boundary or encountering KV cache overflow, the slowdown may be caused by resource bottlenecks rather than reflecting the intrinsic advantage of QeRL. In such cases, restricting the comparison to a single GPU may not represent a realistic or fair setting, since practitioners would not normally train 32B-scale RL models on a single GPU. A comparison under a configuration where the baseline does not suffer from KV cache exhaustion would provide a more accurate and meaningful assessment of the true speedup.

7. **Typos / Minor Issues**

   L235–236: quotation formatting should be corrected (''flatter'' → ``flatter''). The same issue also appears on **L238–239** for ''optimal''.

---

> ### Author Response · Authors · 2025-11-21
>
> **Dear Reviewer 9ptM,**
>
> Thank you for taking the time to provide your valuable and professional suggestions on our paper. We will address each of your questions one by one.
>
> **Q1:** “Baseline consistency in efficiency claims...”
>
> **A1:**  We need to highlight that, both our speedup and memory are compared with **BF16 LoRA and QLoRA (as shown in Table 3), but not memory vs BF16 Full**.  Following your suggestion, we will make the statement more clear. In the abstract, Section 4.3, and the conclusion, we will revise the text to claims of the form:
>
> > “QeRL achieves around a 1.2×–1.5× speedup compared to BF16 LoRA in end-to-end RL training while drastically reducing memory usage, and a 1.5×–2.0× speedup compared to QLoRA.”
> >
>
> We will ensure that each efficiency claim clearly names its corresponding baseline (BF16 Full, BF16 LoRA, or QLoRA). We revise all these statement in the revision manuscript.
>
> **Q2:** “AQN ablation to validate its general effectiveness...”
>
> **A2:** Nice suggestion! AQN was originally motivated as a way to complement quantization noise, but we agree it is important to test its generality. We have added experiments with AQN under full-parameter RL and additional quantization formats (e.g., NF4, MXFP4) with LoRA.
>
> | Model | Training | GSM8K |
> | --- | --- | --- |
> | Qwen2.5-7B-Instruct | - | 76.3 |
> |  | Full-Parameter Tuning | 91.2 |
> |  | Full-Parameter Tuning + AQN | 91.4 |
> |  | BF16 LoRA | 88.1 |
> |  | BF16 LoRA + AQN | 89.7 |
> |  | MXFP4 LoRA | 86.4 |
> |  | MXFP4 LoRA + AQN | 88.1 |
> |  | NF4 LoRA | 85.0 |
> |  | NF4 LoRA + AQN | 85.7 |
>
> Across full-parameter tuning and different quantization formats (NF4, MXFP4, BF16) with LoRA, adding AQN consistently gives modest gains, which is consistent with parameter-noise based exploration in [1,2]. We also observe that the relative improvement is larger in the LoRA setting. Our interpretation is that full-parameter RL can already fit the data well, leaving less headroom for additional exploration, whereas in low-rank (LoRA) updates, AQN-induced exploration provides a stronger benefit.
>
> We will add these results and the above discussion to the revised manuscript to better support the independent contribution and generality of AQN.
>
> [1] Parameter space noise for exploration. ICLR 2018
>
> [2] Noisy networks for exploration. ICLR 2018
>
> **Q3:** “Justification for “Quantization Encourages Exploration””
>
> **A3:**
>
> - **Evidence that quantization itself increases entropy**
>
>     As shown in **Section 3.2 and Figure 5**, the NVFP4 model has a noticeably higher output entropy than the BF16 model from the very beginning of training, and it maintains a relatively high entropy throughout training.
>
>   This behavior is consistent with prior quantization studies [1, 2, 3], where the perplexity of quantized LLMs often increases. Since perplexity and entropy follow the same trend, this suggests that quantization typically raises output entropy. In standard supervised tasks, such entropy increase is usually considered harmful; however, for reasoning‑oriented RL, the higher entropy induced by quantization appears to facilitate exploration and leads to better performance on reasoning tasks.
>
> - **Why we do not try “Quantized Full RL”**
>
>     In fully quantized RL training, one still needs high‑precision master weights and uses low‑precision GEMM only for efficient computation. This means the full model must still be stored in at least 16‑bit on GPU, which requires substantial memory. By contrast, QeRL keeps the static model in 4‑bit, leading to much lower memory usage.
>
>     In FP8 LLM training, there is a known mismatch between low‑precision forward/backward computation and higher‑precision weight updates. **This mismatch makes stable training difficult, and RL is even more sensitive to such precision misalignment than SFT [3,4]**. Recent report [3] has shown that pure FP8 RL, due to amplified train–inference precision mismatch, often leads to severe performance degradation or even model collapse, let alone going down to 4‑bit.
>
>     QeRL (and QLoRA‑style methods) avoid this problem: the same quantized representation is used consistently across rollout, forward, and update, with higher‑precision LoRA parameters handling the updates. **As a result, we do not suffer from the train–inference mismatch, while can benefit from exploration led by quantization.**
>
>
> [1] AWQ: Activation-aware Weight Quantization for LLM Compression and Acceleration. MLSys 2024
>
> [2] GPTQ: Accurate Post-training Compression for Generative Pretrained Transformers. ICLR 2023
>
> [2] QLoRA: Efficient Finetuning of Quantized LLMs. NeurIPS 2023
>
> [3] SGLang RL Report. GitHub
>
> [4] Fast RL training with Quantized Rollouts. GitHub

---

> ### Author Response · Authors · 2025-11-21
>
> **Q4:** “Reporting time cost in main results and clarifying baselines for efficiency claims...”
>
> **A4:** We agree that efficiency should be presented together with performance and that baselines must be explicit. We highlight that  **detailed rollout and end-to-end training times are reported** within **Table 3** and **Appendix J (Tables 5,6,7,8,9)** for different model sizes in our manuscript.
>
> We will check all mentions of speedup in the abstract, Section 4.3, and the conclusion to ensure that each claim states its exact baseline (BF16 LoRA or QLoRA). Following your suggestion, we will make the revision below:
>
> - **Section Abstract (Line 20–21):** Experiments demonstrate that QeRL delivers around 1.5× speedup in the rollout phase compared to QLoRA, and around 1.3× speedup compared to BF16 LoRA in 7B model
> - **Section Conclusion (Line 484–485):** “QeRL achieves around a 1.2×–1.5× speedup compared to BF16 LoRA in end-to-end RL training while drastically reducing memory usage, and a 1.5×–2.0× speedup compared to QLoRA.”
>
>
> **Q5:** “End-to-end speed vs rollout-only speed...”
>
> **A5:** Good Point! In the revision, we will report the full breakdown of  RL time, including rollout, reward computation, actor update, model movement, and total time in below tables.
>
> These results show that rollout remains the dominant cost in LLM RL, and that the rollout speedups largely translate into end‑to‑end wall‑clock gains. We will integrate these tables into the main paper and update the discussion accordingly.
>
> | Qwen2.5-3B-Instruct | rollout | calculate reward | logitupdate_actor | move_model | total time |
> | --- | --- | --- | --- | --- | --- |
> | BF16 LoRA | 2.8 | 0.03 | 0.37 | 0.59 | 3.82 |
> | QLoRA | 5.3 | 0.04 | 0.61 | 0.14 | 6.07 |
> | NVFP4 | 2.5 | 0.03 | 0.53 | 0.17 | 3.21 |
>
> | Qwen2.5-7B-Instruct | rollout | calculate reward | update_actor | move_model | total time |
> | --- | --- | --- | --- | --- | --- |
> | BF16 LoRA | 6.28 | 0.05 | 0.3 | 0.57 | 7.20 |
> | QLoRA | 9.48 | 0.06 | 0.74 | 0.15 | 10.43 |
> | NVFP4 | 4.00 | 0.05 | 0.53 | 0.14 | 4.75 |
>
> | Qwen2.5-14B-Instruct | rollout | calculate reward | update_actor | move_model | total time |
> | --- | --- | --- | --- | --- | --- |
> | BF16 LoRA | 10.24 | 0.01 | 0.51 | 0.68 | 11.45 |
> | QLoRA | 12.30 | 0.01 | 1.05 | 0.21 | 13.56 |
> | NVFP4 | 6.62 | 0.01 | 0.96 | 0.27 | 7.85 |
>
> **Q6:** “Clarifying the ~2× speedup for 14B–32B models...”
>
> **A6:** We need to highlight that we carefully **avoided KV‑cache exhaustion issue** in our experiments.
>
> As described in **Table 8**, for the 32B model we use a standard context length (**input 256 tokens, max completion 2048 tokens**). On a single 80 GB H100, the BF16 baseline can still perform rollout for the 32B model without KV‑cache overflow, and we do not observe KV‑related OOM or throttling. The ~2× rollout speedup therefore does not come from cache failure, but from the reduced memory bandwidth pressure of NVFP4 and computational kernel. For large models, inference becomes increasingly memory‑bound rather than compute‑bound, so lowering weight precision reduces VRAM traffic and yields larger speedups. This matches common observations in large‑scale inference [1, 2].
>
> We also note that QeRL can naturally scale to multi‑GPU settings, where its lower memory footprint allows larger batch sizes and longer context lengths per GPU. At the same time, we deliberately emphasize single‑GPU experiments because many researchers and small teams cannot afford large GPU clusters, so efficient training on resource limited environment is a cutting-edge topic [3, 4, 5] in community. An important goal of QeRL is to make LLM RL training **more accessible under such constrained settings**.
>
> [1]AWQ: Activation-aware Weight Quantization for LLM Compression and Acceleration. MLSys 2024
>
> [2] A Survey on Efficient Inference for Large Language Models. 2024
>
> [3] QLoRA: Efficient Finetuning of Quantized LLMs. NeurIPS 2023
>
> [4] A survey of reinforcement learning for large reasoning models. 2025
>
> [5] LoRA: Low-Rank Adaptation of Large Language Models. ICLR 2022
>
> **Q7:** “Typos / Minor Issues...”
>
> **A7:** Thank you for the suggestion! We will correct the quotation formatting on Lines 235–236 and 238–239 (e.g., use `flatter'' and` optimal'').

---

> > ### Comment · Reviewer_9ptM · 2025-11-27
> > **Additional Clarification Requests from Reviewer 9ptM**
> >
> > Thank you for your detailed responses and revisions. I appreciate the effort put into addressing the earlier concerns. After reading the replies carefully, I still find that several key points require further clarification to ensure the claims in the paper are fully supported and the empirical evidence is aligned with the main contributions.
> >
> > 1. **Time breakdown under the actual training setup**
> >
> > The current breakdown is measured under a speed-test setting.
> > Please provide the end-to-end time breakdown under the main RL training configuration used in Tables 1 and 2, so the reported speedup matches the real training pipeline.
> >
> > 2. **AQN generality beyond GSM8K**
> >
> > The AQN ablation is only shown on GSM8K. To support the claim of generality, please include results on at least one reasoning benchmark (MATH500 / AIME24/25 / AMC23).
> >
> > 3. **Clarifying “quantization encourages exploration”**
> >
> > The current evidence supports this only in the quantized-LoRA setting. If quantized full RL is not evaluated, please adjust the claim to “quantization encourages exploration under LoRA-based RL” to avoid overgeneralization.

---

> > > ### Author Response · Authors · 2025-12-01
> > > **Additional Clarification**
> > >
> > > Dear Reviewer 9ptM,
> > >
> > > Thank you for your reply and informative suggestion! We are so pleased that our previous rebuttal has solved most of your questions and concerns. We have been doing our best to expand the experiment in the past few dozen hours. We will reply to your comments one by one below.
> > >
> > > > Reply 1: The current breakdown is measured under a speed-test setting. Please provide the end-to-end time breakdown under the main RL training configuration used in Tables 1 and 2, so the reported speedup matches the real training pipeline.
> > > >
> > >
> > > Thank you for the suggestion. We now report the end-to-end time breakdown under the exact RL training configuration used for Tables 1 and 2, so the speedup reflects the full training pipeline rather than the microbenchmark.
> > >
> > > Training setup:
> > >
> > > - Hardware: 8 × H100 (80 GB) GPUs with data parallelism
> > > - Per-device batch size: 8
> > > - Gradient accumulation steps: 4
> > > - Output max length: 1024 for 3B/7B models (same as **Figure 1**); 4096 for the 14B model
> > >
> > > Under this setting, we provide the per-GPU time breakdown within the parallel QeRL training run. The overall speedup for the full pipeline is already shown in **Figure 1, Table 3, and Appendix J (Tables 5–9)**. The breakdown speed results are aligned with the original Tables in our manuscript.
> > >
> > > | Model | **rollout** | **calculate reward** | **update_actor （include gradient accumulation）** | **move_model** | **total time** |
> > > | --- | --- | --- | --- | --- | --- |
> > > | 3B | 3.28 | 0.04 | 2.47 | 0.48 | 6.27 |
> > > | 7B | 4.62 | 0.04 | 2.56 | 0.67 | 9.49 |
> > > | 14B | 15.32 | 0.02 | 7.38 | 0.95 | 23.67 |
> > >
> > > This table is added to in Table 4 of our revised manuscript.
> > >
> > > > Reply 2: The AQN ablation is only shown on GSM8K. To support the claim of generality, please include results on at least one reasoning benchmark (MATH500 / AIME24/25 / AMC23).
> > > >
> > >
> > > Great point! We agree that additional benchmarks are important to support the generality of AQN. In the revision, we add a new training ablation to further evaluate AQN under different data formats (NVFP4 LoRA, BF16 Full, BF16 LoRA), and report results on MATH500. Specifically, we train 7B models on the BigMath dataset for each configuration, and then evaluate them on MATH500 to measure the impact of AQN beyond GSM8K. Each ablation run requires approximately 50 hours of training. We found that the role of AQN in MATH500 show the same trend as in GSM8K.
> > >
> > > | Model | Training | GSM8K | MATH500 |
> > > | --- | --- | --- | --- |
> > > | Qwen2.5-7B-Instruct | - | 76.3 | 74.8 |
> > > |  | Full-Parameter Tuning | 91.2 | 77.4 |
> > > |  | Full-Parameter Tuning + AQN | 91.4 | 77.2 |
> > > |  | BF16 LoRA | 88.1 | 77.0 |
> > > |  | BF16 LoRA + AQN | 89.7 | 77.8 |
> > > |  | NVFP4 LoRA | 88.5 | 76.8 |
> > > |  | NVFP4 LoRA + AQN | 90.8 | 77.4 |
> > >
> > > This table is added to in Table 7 in Appendix section of our revised manuscript.
> > >
> > > > Reply 3: The current evidence supports this only in the quantized-LoRA setting. If quantized full RL is not evaluated, please adjust the claim to “quantization encourages exploration under LoRA-based RL” to avoid overgeneralization.
> > > >
> > >
> > > Following your suggestion, we have carefully revised the claims in the manuscript. We now restrict our conclusions to the LoRA-based, parameter-efficient setting and use more precise terms such as “LoRA-based RL” instead of making statements about full-parameter quantized RL. The revised statements can be found in our Abstract, Introduction, Method and Conclusion sections in the revision.

---

### Official Review · Reviewer_a31R · 2025-10-31

**Soundness:** 3
**Presentation:** 3
**Contribution:** 4
**Rating:** 8
**Confidence:** 3

**Summary:**

This paper introduces QeRL, a novel framework that integrates NVFP4 quantization with LoRA to enhance the efficiency and effectiveness of RL for LLMs. The work is motivated by the significant computational bottlenecks—high memory usage and slow rollout speeds—inherent in RL training for LLMs. While existing approaches like QLoRA address memory concerns, they often introduce slowdowns and are not designed to leverage the unique dynamics of RL. QeRL's primary innovation lies in its dual approach: it achieves computational efficiency while also demonstrating, for the first time, that the quantization error (or "noise") can be a powerful, implicit mechanism for enhancing exploration in RL.

To overcome the limitation of static quantization noise, the paper further proposes an Adaptive Quantization Noise (AQN) mechanism. AQN dynamically injects and schedules noise, integrated in a zero-overhead manner into LayerNorm layers, to balance exploration and exploitation throughout training.

The experimental validation is extensive and convincing. The authors demonstrate that QeRL achieves over 1.5x speedup in the rollout phase and reduces memory usage by 50-60% compared to 16-bit LoRA, enabling the training of a 32B parameter model on a single H100 GPU. Crucially, QeRL does not sacrifice performance for efficiency; it consistently matches or surpasses the performance of 16-bit LoRA and often reaches the performance ceiling of full-parameter fine-tuning on challenging mathematical reasoning benchmarks like GSM8K and MATH.

**Strengths:**

1. Novel and Counter-Intuitive Core Insight: The most significant strength of this paper is its conceptual leap: reframing quantization error from a detrimental artifact (as it is often viewed in SFT) into a beneficial exploratory force in RL. The careful analysis in Figures 3 and 5, showing the entropy increase and its correlation with improved reward growth, provides solid evidence for this claim.

2. Holistic and Practical Framework (QeRL + AQN): The paper builds a complete, end-to-end framework. The combination of high-performance NVFP4 quantization with LoRA is a pragmatic solution to the memory and speed problem. The introduction of the Adaptive Quantization Noise (AQN) mechanism is a clever innovation that addresses the inherent limitation of static noise. Significantly, the noise vector can be elegantly merged into LayerNorm to avoid extra computational cost. This demonstrates a deep understanding of both algorithmic needs and system-level constraints.

3. Extensive and Compelling Empirical Evaluation: The authors conduct a thorough ablation study across multiple model sizes, two  RL algorithms (GRPO and DAPO), and several challenging mathematical benchmarks (GSM8K, BigMath). On the efficiency front, QeRL achieves substantial memory reduction, shrinking model sizes to 25-30% of their 16-bit counterparts, while also providing a 1.2x to 1.5x end-to-end training speedup and a greater than 2x speedup in the rollout phase for larger models. Crucially, these efficiency gains do not come at the cost of performance; QeRL consistently achieves superior or comparable final accuracy and faster reward convergence compared to 16-bit LoRA and QLoRA, often matching the high-performance benchmark set by full-parameter fine-tuning.

**Weaknesses:**

1. Limited Task and Domain Generalization: The empirical validation, while extensive, is heavily focused on mathematical reasoning tasks (GSM8K, MATH, AIME, AMC). While these are excellent proxies for complex, multi-step reasoning, it remains an open question whether the benefits of QeRL generalize equally well to other domains such as coding.

2. Dependency on Hardware and Kernels: The significant speedups of QeRL are contingent on the efficient NVFP4 format and the supporting Marlin kernel, which are tied to recent NVIDIA architectures (Hopper, Blackwell). This might limit immediate adoption for researchers without access to the latest hardware. A brief discussion on the performance or feasibility on more widely available hardware (e.g., Ampere) would be helpful for context.

**Questions:**

1. The exponential scheduler for AQN was chosen after comparing several options. Could you elaborate on the sensitivity of the performance to the key hyperparameters of AQN, such as $\sigma_{start}$ and $\sigma_{end}$? Was the choice of 10 evenly spaced intervals for noise adjustment based on empirical tuning, or is there a heuristic for setting this based on the total number of training steps?

2. While the paper provides a strong empirical and intuitive explanation for how quantization noise increases entropy and aids exploration, could you elaborate on the theoretical connection between the specific properties of NVFP4 quantization noise (e.g., its distribution and magnitude) and established RL exploration theory? A deeper formal analysis could further solidify the foundational contribution of this work.

---

> ### Author Response · Authors · 2025-11-21
>
> Dear Reviewer a31R,
>
> Thank you for highlighting and appreciating our contributions in exploring the quantization-enhanced RL! Below, we will address your questions one by one.
>
> **Q1:** “Limited Task and Domain Generalization...”
>
> **A1:** Thank you for your suggestion! Here, we extended our experiments beyond math reasoning. We added results on more generalized language task **CommQA (dialogue) and SafeRLHF (safety‑critical RL)**.  For code tasks, RL is usually applied after supervised fine-tuning, and requires an external compiler or executor as a reward oracle. This makes training and evaluation much more time‑consuming, which we could not complete within the current discussion and revision cycle. We therefore leave a systematic study of code RL to future work.
>
> **Safety-Critical RL**
>
> | Model | Training | Memory (at least) | PKU-SafeRLHF-QA |
> | --- | --- | --- | --- |
> | Qwen2.5-3B-Instruct | - | - | 78.4 |
> | Qwen2.5-3B-Instruct | Full-Parameter Tuning | > 40 GB | 87.2 |
> | Qwen2.5-3B-Instruct | BF16 LoRA | > 25 GB | 85.5 |
> | Qwen2.5-3B-Instruct | QeRL | > 9 GB | 86.3 |
>
> **Dialogue RL**
>
> | Model | Training | Memory (at least) | Commonsense_QA |
> | --- | --- | --- | --- |
> | Qwen2.5-3B-Instruct | - | - | 66.9 |
> | Qwen2.5-3B-Instruct | Full-Parameter Tuning | > 40 GB | 79.2 |
> | Qwen2.5-3B-Instruct | BF16 LoRA | > 25 GB | 72.5 |
> | Qwen2.5-3B-Instruct | QeRL | > 9 GB | 71.0 |
>
> **Q2:** “Dependency on Hardware and Kernels...”
>
> **A2:** We agree that our largest speedups rely on NVFP4 and the Marlin kernel, which currently require recent NVIDIA architectures (Hopper, Blackwell). However, the main contribution of QeRL is to show the **benefits of exploration for RL training**, not only the specific speedups.
>
> On other GPU architectures such as Ampere, one can still apply QeRL using slower de-quantization schemes but not NVFP4 * BF16 kernel. In that setting, the throughput gains are smaller, but we still observe advantages from the reduced memory footprint and improved exploration behavior brought by QeRL. We will add a short discussion of this feasibility on Ampere-class hardware in the revised manuscript, as you suggest.
>
> **Q3:** “Could you elaborate on the sensitivity of the performance to the key hyperparameters of AQN”
>
> **A3:** We observe that larger models are more sensitive to weight noise, so we use relatively small noise levels and avoid aggressive settings.
>
> Concretely, we use start noise levels in the range 1e-1 ~1e-2 and end noise levels in the range 1e-3 ~ 5e-4. These are typical scales in traditional RL, where they provide stable exploration without strongly disrupting the policy. We observe similar behavior in LLM RL: within this range, training remains stable and performance does not change sharply. To make this more quantitative, we added a small sensitivity study on GSM8K with Qwen2.5‑3B‑Instruct.
>
> These results show that AQN is reasonably robust within this band, with all settings giving stable training and similar performance. We also notice that too large noise (above 1e-1) can start to hurt performance, so we keep the start noise at or below 1e-1 in all our main experiments.
>
> We will include this table and a brief sensitivity discussion in the revised appendix.
>
> | Model | start | end | GSM8K |
> | --- | --- | --- | --- |
> | Qwen2.5-3B-Instruct | 1e-1 | 1e-3 | 82.5 |
> |  | 1e-2 | 1e-3 | 83.7 |
> |  | 1e-1 | 5e-4 | 83.1 |
> |  | 1e-2 | 5e-4 | 83.7 |

---

> ### Author Response · Authors · 2025-11-21
>
> **Q4:** “A deeper formal analysis could further solidify the foundational contribution of this work.”
>
> **A4:** Thank you for the good suggestion. Prior works [1,2] have shown that pre-trained weights in large neural networks are approximately Gaussian or Laplace distributed. NVFP4 quantization is designed to approximate this bell-shaped distribution, so the quantized weights are also near-Gaussian. The quantization process can therefore be viewed as mapping one near-Gaussian weight distribution to another, and the mismatch appears as additive quantization noise. In the idealized case where both are Gaussian, this noise is itself approximately Gaussian, i.e., it acts as small, near-isotropic perturbations to the parameters. This is closely aligned with parameter-space noise in RL [3,4], where injecting small random perturbations into model parameters is known to improve exploration by diversifying the policy’s outputs.
>
> As we show in Section 3.2, this quantization noise in LLMs directly affects the post-softmax next-token distribution, making it flatter and increasing output entropy:
>
> $$
> H(P) = -\sum_{i=1}^{V} P_i \text{log} P_i
> $$
>
> $$
> P(y = i|x) = softmax(z)_i = \frac{\text{exp}(z_i)}{\sum _{j=1}^{V}exp(z_j)}
> $$
>
> In the original model, the output features of the last few LLM layers tend to have almost fixed large activations, i.e., the distribution of $z$ is very sharp. Under weight quantization noise, the distribution of $z$ becomes flatter, and the variance of the probability vector $P$ increases. Consequently, we obtain: $[H(P_{quant})] \ge H(P)$. **Figure 5 further illustrates that higher entropy improves exploration in LLM-based RL**, consistent with our analysis.
>
> [1] Post-training piecewise linear quantization for deep neural networks. ECCV 2020
>
> [2] Pushing the Limit of Post-Training Quantization for LLMs. ICML 2024
>
> [3] Parameter space noise for exploration. ICLR 2018
>
> [4] Noisy networks for exploration. ICLR 2018

---

### Official Review · Reviewer_g8xh · 2025-11-02

**Soundness:** 3
**Presentation:** 3
**Contribution:** 4
**Rating:** 6
**Confidence:** 4

**Summary:**

This paper introduces QeRL, a framework that combines NVFP4 quantization with Low-Rank Adaptation (LoRA) for efficient training of large language models in reinforcement learning. The key insight is that quantization noise can enhance exploration during RL (unlike in supervised fine-tuning, where it's detrimental), leading to both improved efficiency and better performance. The authors propose an Adaptive Quantization Noise (AQN) mechanism that dynamically adjusts noise throughout training. Experiments on mathematical reasoning benchmarks (GSM8K, MATH500, AIME, AMC) demonstrate a 1.5× speedup in the rollout phase, the ability to train 32B models on a single H100 GPU, and performance matching or exceeding that of full-parameter fine-tuning on 7B models.

**Strengths:**

Practical impact: Enables RL training of large models (up to 32 B) on a single GPU with significant speedups.
 Exploration insight: Demonstrates that adaptive quantization noise can raise policy entropy and aid reward learning.
Strong empirical results: Matches or exceeds 16-bit LoRA / QLoRA on math reasoning tasks.
Good ablation coverage: Studies noise schedules and LoRA ranks; efficient integration using Marlin kernels.

**Weaknesses:**

Positioning and comparison to recent quantized RL and post-training quantization work: While the paper surveys relevant literature, it does not cite or compare with several recent approaches critical to advances in quantized RL or LLM quantization. For instance, Quantized Reinforcement Learning (QuaRL) [Krishnan et al., 2020, 2022] and newer post-training quantization frameworks (RPTQ, VPTQ, LRQuant, PREFIXQUANT, QuIP$). This makes it challenging to position the core empirical claims and efficiency gains in relation to the current state-of-the-art.

Limited statistical rigor: No multi-seed averages or error bars; robustness to randomness is unknown.

Scope restricted to math reasoning: No results for code, dialogue, or safety-critical RL tasks.

Missing FlashRL baseline: Discussed in related work but not compared directly.

Hyperparameter fairness: Learning-rate differences may partly explain improved stability; further tuning of BF16 LoRA could close the gap.

Hardware specificity: To the best of my knowledge, NVFP4 / Marlin optimizations are NVIDIA-centric; portability to other accelerators is unclear.

Potential overstatement of claims: In the introduction and conclusion, there are hints of broad generalization (e.g., “our technique can be seamlessly adapted to richer…datasets”, p. 21) without empirical demonstration. While acknowledging the focus on math reasoning, a slightly more tempered framing would match the current evidence base.

**Questions:**

Could you provide variance/error bars for multiple runs of GSM8K/MATH500?

Can you include a direct comparison with FlashRL to quantify gains beyond precision-mismatch correction?

How sensitive is AQN to the start/end noise levels and decay schedule?

Did you try stabilizing BF16 LoRA with smaller ranks or gradient clipping so it can use 1e-5 LR?

Please release a concise 32B single-GPU recipe (context length, microbatching, KV cache type, LoRA rank, and optimizer).

Any early results on code or safety-oriented tasks to test AQN’s generalization? Can the authors provide more quantitative discussion about the failure modes of AQN or when exploration becomes detrimental, e.g., in very large models or with more complex tasks?

---

> ### Author Response · Authors · 2025-11-21
>
> **Dear Reviewer g8xh,**
>
> We sincerely thank you for the in-depth feedback! We highly value each of your comments, and all your concerns are addressed point by point.
>
>
> **Q1:** “Positioning and comparison to recent quantized RL and post-training quantization work...”
>
> **A1:** Thank you for this suggestion. In the revision, we will add discussion in related works for QuaRL and recent post‑training quantization methods (RPTQ, VPTQ, LRQuant, PREFIXQUANT, QuIP#), and more clearly position our contributions relative to these works. The design of advanced PTQ optimization strategies is not the main focus of our work. Our goal is to study how quantized models can directly benefit LLM RL training, rather than to propose a new quantization algorithm. We also observed that many of these PTQ methods rely on vector quantization, which currently makes it non‑trivial to combine them cleanly with LoRA during training and to realize efficient inference speedups; we leave a more systematic integration as future work.
>
> QuaRL mainly targets RL for small models. Its usage is conceptually similar to FlashRL in the LLM setting, where a quantized model is used for rollout but full‑precision weights are still needed for policy optimization. This leads to a significant **train–inference mismatch** and a **training memory cost** that remains close to full‑parameter fine‑tuning. By contrast, QeRL keeps the training and inference models aligned and thus avoids this mismatch, while achieving substantially lower memory consumption during RL training.
>
> **Q2:** “No multi-seed averages or error bars; robustness to randomness is unknown....”
>
> **A2:** Good point！Thank you for the suggestion. As mentioned in **Section 4.1**, we already use multiple runs to obtain more robust evaluation results. We have now computed and reported the mean and standard deviation over multiple runs for GSM8K and MATH500 of 3B and 7B model, and we include these results in the following tables.
>
> | Model | Training | GSM8K |
> | --- | --- | --- |
> | Qwen2.5-3B-Instruct | - | 61.2 (*± 1.9*) |
> |  | Full-Parameter Tuning | 84.4 (*± 1.5*) |
> |  | BF16 LoRA | 76.1 (*± 1.6*) |
> |  | QeRL | 83.7 (*± 1.1*) |
>
> | Model | Training | GSM8K | MATH 500 |
> | --- | --- | --- | --- |
> | Qwen2.5-7B-Instruct | - | 76.3 (*± 1.2*) | 78.4 (*± 1.0*) |
> |  | Full-Parameter Tuning | 91.2  (*± 0.6*) | 77.4 (*± 0.4*) |
> |  | BF16 LoRA | 88.1  (*± 0.9*) | 77.0 (*± 0.7*) |
> |  | QeRL | 90.8  (*± 1.2*) | 77.4 (*± 1.0*) |
>
> **Q3:** “Scope restricted to math reasoning...”
>
> **A3:** Following your nice suggestion, we  extended our experiments beyond math reasoning. We added results on CommQA (dialogue) and SafeRLHF (safety‑critical RL) to examine other RL settings.
>
> We observe that safety‑critical tasks often require careful step‑by‑step analysis to produce safe and accurate responses, so they benefit from reasoning‑enhanced RL. In this regime, QeRL consistently outperforms BF16 LoRA, similar to the gains seen on math benchmarks. In contrast, CommonsenseQA relies more on knowledge recall than on complex reasoning, so the benefit from exploration and reasoning‑focused RL is less pronounced. We will describe these observations briefly in the revised discussion section. Our current goal is to study RL on purely base models. For code tasks, RL is usually applied after supervised fine-tuning, and requires an external compiler or executor as a reward oracle. This makes training and evaluation much more time‑consuming. We therefore leave a systematic study of code RL to future work.
>
> | Model | Training | Memory (at least) | PKU-SafeRLHF-QA |
> | --- | --- | --- | --- |
> | Qwen2.5-3B-Instruct | - | - | 78.4 |
> |  | Full-Parameter Tuning | > 40 GB | 87.2 |
> |  | BF16 LoRA | > 25 GB | 83.5 |
> |  | QeRL | > 9 GB | 85.3 |
>
> **Dialogue RL**
>
> | Model | Training | Memory (at least) | Commonsense_QA |
> | --- | --- | --- | --- |
> | Qwen2.5-3B-Instruct | - | - | 66.9 |
> |  | Full-Parameter Tuning | > 40 GB | 79.2 |
> |  | BF16 LoRA | > 25 GB | 73.5 |
> |  | QeRL | > 9 GB | 73.2 |
>
>
> **Q4:** “Missing FlashRL baseline...”
>
> **A4:** Thank you for this helpful comments. Following your comment, we have added a direct comparison between QeRL and FlashRL on 7B model in the revised version.
>
> FlashRL uses an 8‑bit model for rollout and a 16‑bit model for training. Its main goal is to speed up training while keeping a full‑precision training model, so the memory cost during training remains similar to **full‑parameter fine‑tuning and it cannot run RL on a single GPU in our setting**. In the new experiments, we report both performance and resource consumption for QeRL and FlashRL, and we briefly discuss the gains that go beyond correcting the train–inference precision mismatch.
>
> | Model | Training | Rollout bit-width | Memory (at least) | GSM8K |
> | --- | --- | --- | --- | --- |
> | Qwen2.5-7B-Instruct | FlashRL (w TIS) | 8-bit | > 70 GB | 89.2 |
> |  | QeRL | 4-bit | > 15 GB | 90.8 |

---

> ### Author Response · Authors · 2025-11-21
>
> **Q5:** “Learning-rate differences may partly explain improved stability...”
>
> **A5:** As described in **Appendix E and I**, we tune learning rate and LoRA rank for both methods under the same protocol, and we use the same gradient clipping range for QeRL and BF16 LoRA to ensure a fair and stable comparison. Empirically, BF16 LoRA is most stable around a learning rate of 5e-6; under this setting QeRL still achieves higher reward and more stable training curves.
>
> We also tested multiple LoRA ranks and observed the same trend: QeRL remains stable and performs well, while BF16 LoRA tends to show instability across settings. When we increase the learning rate toward 1e-5, QeRL benefits from the larger step updates, but BF16 LoRA becomes even more unstable despite gradient clipping and smaller ranks.
>
> **Q6:** “Hardware specificity”
>
> **A6:** We agree that our current implementation is NVIDIA‑centric: QeRL is built on NVFP4 / Marlin kernels, which are specifically optimized for NVIDIA GPUs, and we currently do not have access to non‑NVIDIA accelerators. Our main claim, however, is about the **algorithmic benefit of combining quantization with LoRA for RL**, rather than about a particular hardware backend.
>
> Thank you for the suggestion. We also note that low‑bit formats such as NVFP4 and other inference kernels can in principle be supported by other platforms as corresponding operators and inference kernels become available. We expect future 4‑bit operator support on other hardware to make QeRL portable with similar benefits, and we will add a brief discussion of this point in the main text.
>
> **Q7:** “In the introduction and conclusion, there are hints of broad generalization...”
>
> **A7:** Thank you for this careful observation. Following your nice suggestion, we have also extended our experiments beyond math reasoning and added results on **CommonsenseQA (dialogue‑style QA)** and **SafeRLHF (safety‑critical RL)**, which are reported and briefly discussed in the revised version (please refer to Q3). We also highlight that the efficiency benefit of QeRL (**low memory and fast single‑GPU training**) comes from the training scheme rather than from properties of a specific dataset, and thus is expected to transfer across tasks.
>
> **Q8:** “Could you provide variance/error bars for multiple runs of GSM8K/MATH500?”
>
> **A8:** Please refer to Q2.
>
> **Q9:** “Can you include a direct comparison with FlashRL to quantify gains beyond precision-mismatch correction?”
>
> **A9:** Please refer to Q4.
>
> **Q10:** “How sensitive is AQN to the start/end noise levels and decay schedule?”
>
> **A10:** Our experiments suggest that AQN is reasonably robust to the choice of start/end noise levels and the decay schedule, as long as they stay in a standard RL range.
>
> Concretely, we use start noise levels in the range 1e-1 ~1e-2 and end noise levels in the range 1e-3 ~ 5e-4. These are typical scales in traditional RL, where they provide stable exploration without strongly disrupting the policy. To make this more quantitative, we added a small sensitivity study on GSM8K with Qwen2.5‑3B‑Instruct.
>
> These results show that AQN is reasonably robust within this band, with all settings giving stable training and similar performance. We also notice that too large noise (above 1e-1) can start to hurt performance, so we keep the start noise at or below 1e-1 in all our main experiments. We will include this table and a brief sensitivity discussion in the revised appendix.
>
> | Model | start | end | GSM8K |
> | --- | --- | --- | --- |
> | Qwen2.5-3B-Instruct | 1e-1 | 1e-3 | 82.5 |
> |  | 1e-2 | 1e-3 | 83.7 |
> |  | 1e-1 | 5e-4 | 83.1 |
> |  | 1e-2 | 5e-4 | 83.7 |
>
> **Q11:** “Did you try stabilizing BF16 LoRA with smaller ranks or gradient clipping so it can use 1e-5 LR?”
>
> **A11:** Please refer to Q5.
>
>
> **Q12:** “Please release a concise 32B single-GPU recipe (context length, microbatching, KV cache type, LoRA rank, and optimizer).”
>
> **A12:** Thank you for the suggestion. Appendix E already describes our training hyperparameters; following your request, we now provide a concise 32B single‑GPU recipe. And all the code and training recipes will be open-sourced! In 32B single‑GPU experiments we use (delete redundant detail):
>
>
> ```bash
> perdevice_train_batch_size=8
> gradient_accumulation_steps=32
> lora_rank=16
> lora_alpha=16
> DATA_NAME="gsm8k" # dapomath, gsm8k, codeforces
> optim="adamw_8bit" # Optimizer type, adamw_8bit or adamw
>
> sigma_start=1e-2
> sigma_end=1e-4
>
> if [ "$DATA_NAME" == "gsm8k" ]; then
> max_prompt_length=256
> max_completion_length=2048
> max_seq_length=2500
> elif [ "$DATA_NAME" == "bigmath" ]; then
> max_prompt_length=1024
> max_completion_length=4096
> max_seq_length=5500
> fi
> rl_mode="dapo"
> beta=0.0
> epsilon_high=0.28
> lr=1e-5
> --adam-beta1 0.9
> --adam-beta2 0.99
> --weight-decay 0.1
> --warmup-ratio 0.1
> --lr-scheduler-type "cosine"
> --num-train-epochs 1
> ```

---

> ### Author Response · Authors · 2025-11-21
>
> **Q13:** “Any early results on code or safety-oriented tasks to test AQN’s generalization?”
>
> **A13:**  We have added experiments on a safety‑oriented RL task (**SafeRLHF, please refer to Q3**), and will report the results in the revised paper. We observe that safety‑critical tasks often require careful, step‑by‑step reasoning to produce safe and accurate answers, and in this setting QeRL brings clear gains, consistent with our math‑reasoning results.
>
> Regarding failure modes and sensitivity, we see two main patterns:
>
> 1. **Larger models:**
>
>     For larger models, we find that weights are more sensitive to injected noise. In practice, we therefore use smaller noise scales (both start and end values) and avoid aggressive settings. With such conservative noise, training remains stable.
>
> 2. **When exploration becomes detrimental:**
>
>     We note that the benefit of quantization noise is largest during the early phase when the reward improves quickly. After this phase, the marginal gain from additional noise becomes small, and excessive noise can slightly hurt performance. Based on this, we recommend:
>
>     - using higher noise only in the early, fast‑improvement stage to encourage exploration;
>     - decaying to relatively small noise levels in the later stage to avoid unnecessary degradation.

---

### Author Response · Authors · 2025-11-21
**General Response**

Dear Reviewers and ACs:

Thank you very much for your constructive comments and insightful reviews which help improve our previous manuscript. We have carefully taken all the suggestions and made detailed revision to our previous draft, with the main changes marked blue in the draft.

Specifically, we have made the following improvement:

In the main paper:

1. Clarify the speedup comparison with both QLoRA and BF16 LoRA
2. Add the discussion on FlashRL and QuaRL.
3. Add a breakdown RL training time in Table 4.
4. Add more generalized RL benchmarks (SafeRLHF, Commonsense).

In the Appendix:

1. Add discussion on other PTQ methods.
2. Add an detailed experiments of FlashRL and QeRL in Table 13.
3. Add more ablation of AQN on different precision training setting in Table 7.
4. Add a comparison of different noise scale in Table 8.

We again sincerely thank all the reviews and ACs’ effort on our submission!

Best regards, Paper 2163 Authors

---

### Comment · Area_Chair_YNmA · 2025-11-27
**Follow-up on Discussion**

Dear Reviewers,

The authors have submitted their rebuttal for paper. Please check the responses and confirm whether your questions are resolved.
Best,
AC

---

### Meta-Review · Area_Chair_Gd9j · 2026-01-06

**Summary:**

The final decision for this submission is Accept. The reviewers reached a consensus that the paper presents a valuable contribution to efficient Reinforcement Learning (RL) by enabling single-GPU training for large models and offering a novel insight regarding quantization noise as a mechanism for exploration. The submission was praised for its strong empirical gains, clear write-up, and the practical utility of the proposed framework combining NVFP4, LoRA, and Adaptive Quantization Noise (AQN). While the initial scores reflected general enthusiasm, the primary barrier preventing a higher unanimous endorsement was the positioning of the work against recent quantized RL and Post-Training Quantization (PTQ) baselines, alongside concerns regarding the generalizability of the approach beyond mathematical reasoning tasks and specific NVIDIA hardware architectures.

**Reviewer Concerns:**

The authors provided a comprehensive rebuttal that successfully improved the completeness of the work by addressing the majority of empirical and methodological concerns. Specifically, the addition of the FlashRL baseline comparison (7B), including memory metrics, significantly strengthened the efficiency claims. Furthermore, the expansion of the evaluation suite to include SafeRLHF and CommonsenseQA demonstrated utility beyond the initial math-focused scope, while the inclusion of multi-seed mean and standard deviation reporting satisfied the requirements for statistical rigor. The authors also clarified the sensitivity of AQN hyperparameters and successfully argued for the algorithmic portability of the method, even if current implementations are kernel-specific. However, questions regarding the soundness of the method's theoretical underpinnings and universal hardware applicability remain partially outstanding. While the informal linkage between Gaussian-like noise and entropy increase offers intuition, a formal theoretical framework connecting NVFP4 noise to RL theory is still light. Additionally, the dependency on NVIDIA-centric kernels implies that portability beyond this specific hardware ecosystem is theoretically possible but practically unproven in the current manuscript. Despite these residual limitations, the core contributions regarding efficiency and the exploration hypothesis are well-supported.

**Reviewer Scores:**

Based on the substantial improvements made during the rebuttal phase, the reviewers' scores are projected to evolve positively. Reviewer g8xh, initially a 6, would likely move to an 8, as their primary concerns regarding positioning against QuaRL/RPTQ, statistical rigor, and the missing FlashRL baseline were directly and effectively addressed, leaving only minor concerns about code task generalization. Reviewer a31R, who already assigned an 8, would maintain this strong score; their critiques regarding domain generalization and parameter sensitivity were mitigated by the additional experiments, rendering the remaining theoretical lightness acceptable for an empirical track. Reviewer 9ptM is projected to increase from a 4 to a 6, as the authors resolved the major clarity issues surrounding mixed baselines and rollout versus end-to-end time breakdowns, effectively scoping the claims to LoRA-based RL to avoid overgeneralization. Finally, Reviewer dYgm would likely upgrade from a 6 to an 8, given that the addition of new domains and the quantification of AQN improvements clarified the initial doubts regarding the modest appearance of gains in the smoothed plots.

---

### Decision · Program_Chairs · 2026-01-26

Accept (Poster)